

# The influence of anthropogenic climate change on Super Typhoon Odette (Typhoon Rai) and its impacts in the Philippines

Ben Clarke[1], Sihan Li[2], Ralf Toumi[3], Nathan Sparks[3]

[1]Centre for Environmental Policy, Imperial College London, UK
[2]School of Geography and Planning, The University of Sheffield, UK
[3]Grantham Institute - Climate Change and the Environment, Imperial College London, UK

*Correspondence to*: Ben Clarke (b.clarke@imperial.ac.uk)



**Abstract.** Super Typhoon Odette (Typhoon Rai) made landfall in the Philippines as a category 5 tropical cyclone on 16th December 2021. It brought the compounding effects of extreme rainfall, high winds and storm surge to large parts of the southern-central Philippines, particularly Cebu and Bohol. It was the second costliest typhoon on record for the Philippines up until 2021, causing nearly a billion dollars (US) in direct damage and widespread disruption. In this study, the extreme rainfall and high winds observed during this storm are assessed to determine the influence of anthropogenic climate change (ACC),

using three different methods, which focus on the circulation patterns, high rainfall and strong winds associated with Odette, respectively. First, we check that the current generation of higher resolution models used in attribution studies can capture the low sea level pressure anomaly associated with Typhoon Odette and hence can be used to study this type of event. A short analysis then compares such circulation analogues and the associated meteorological extremes over three time periods: past (1950-1970), contemporary (2001-2021), and future (2030-2050). Second, a multi-method multi-model probabilistic event

attribution finds that extreme daily rainfall such as that observed during Typhoon Odette has become about twice as likely during the Typhoon season over the southern-central Philippines due to ACC. Third, a large ensemble tropical cyclone hazard model finds that the wind speeds of category 5 landfalling typhoons like Odette have become approximately 70% more likely due to ACC. The combined results show that both extreme rainfall and wind speeds in the Philippines due to storms like Odette have become significantly more likely and intense due to ACC. Based on these results and compound event attribution theory,

we further conclude that ACC has likely more than doubled the likelihood of a compound event like Typhoon Odette. Finally, while the impacts were caused by a range of factors including exposure and vulnerability, ACC played a significant role in amplifying the damage, and this risk will very likely continue to grow with increasing levels of warming.





# 1 Introduction

## 1.1 Super Typhoon Odette

Super Typhoon Odette first formed on Dec 10[th] as a tropical depression over the western Pacific. It moved WNW and entered the Philippines' area of responsibility on the 14[th], becoming officially named Odette ('Rai' internationally) (ReliefWeb, 2025).

It then rapidly intensified on the 15[th] and morning of the 16[th] making multiple landfalls (when the surface centre of the storm passes over the coastline) as a category 5 super typhoon throughout the Visayas on the afternoon and evening of the 16[th] (Chan et al., 2022). It progressed across the Visayas from east to west, bringing heavy rainfall, high winds, with maximum sustained wind speed of 175 km/h and gusts of up to 240 km/h (ReliefWeb, 2025), and storm surges (abnormal rises in sea level due to the typhoon) of several metres (Esteban et al., 2023). The transit of the storm resulted in exceptionally disastrous impacts,

especially in the provinces of Cebu and Bohol, in which storm surges reached 2.5m and 4m, respectively (Esteban et al., 2023).

The compounding nature of multiple meteorological hazards (heavy precipitation, strong winds, storm surge) from the tropical storm led to flooding, both coastal and rainfall-driven, and the destruction of property, agriculture and infrastructures. 2.1 million homes were damaged, with 425000 considered destroyed entirely (ReliefWeb, 2025). The damages totalled approximately US$915 million, 86000 ha of agricultural land was damaged, and over 300 people were killed (EMDAT, 2019;

Mata et al., 2023), altogether making it the second costliest typhoon to strike the Philippines on record at the time of occurrence after Typhoon Haiyan in 2013. When Typhoon Megi/Agaton struck the Philippines in April 2022, it affected some of the same areas in Leyte province affected by Odette, which compounded impacts further (ReliefWeb, 2025).

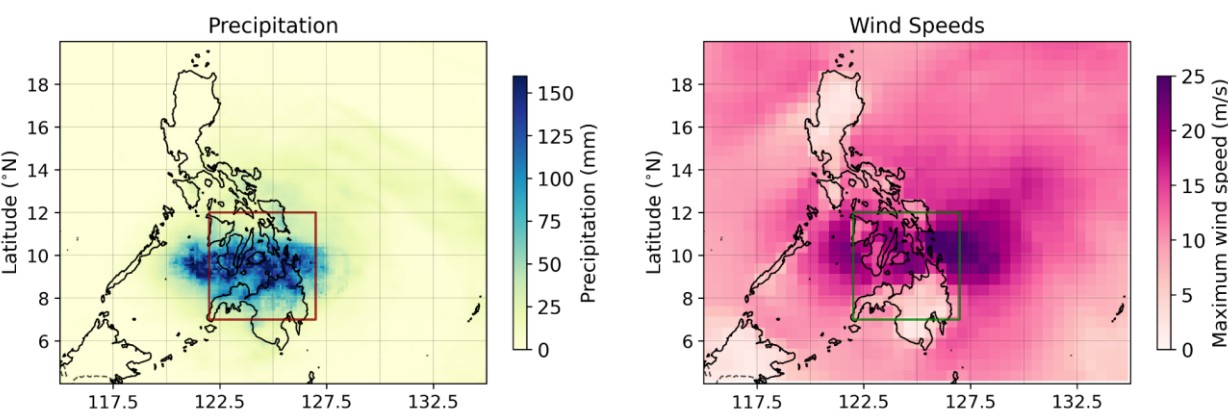

*Figure 1: Daily precipitation (mm/day) and maximum wind speeds (m/s) observed on the 16th of December 2021, during the*

*passage of Typhoon Odette over the Philippines. The study region used in Sect. 2 is highlighted in red (left) and green*
   *(right). Precipitation data is from the Multi-Source Weighted-Ensemble Precipitation and daily maximum wind speeds are*





*from ERA5 (see Sect. 3.2.1). Note: reanalysis datasets struggle to capture the extreme winds experienced during category 4 and 5 tropical cyclones, so the values in the right-hand figure are purely illustrative.*

**1.2 Climate change and tropical cyclones in the Philippines**

In 2021, the concentration of carbon dioxide in the atmosphere was 417 parts-per-million (ppm), approximately 50% above preindustrial levels, and the global mean surface temperature was approximately 1.1 °C above the preindustrial baseline (Masson-Delmotte et al., 2021). Both local and global climates are affected by warming, caused primarily by the burning of fossil fuels and other human activities. Changing patterns of extreme weather are one of the principal near-term effects of anthropogenic climate change on societies (Seneviratne et al., 2021).

The Philippines regularly ranks amongst the nations most affected by climate-related extremes (Eckstein et al., 2021). On average, approximately 19 TCs enter the Philippines 'Area of Responsibility' every year, and 9 TCs make landfall, more than any other nation (Cinco et al., 2016; Santos, 2021). The peak Typhoon season for the Philippines is from June-December, with the fewest experienced around February-March, though they have historically occurred in all months (Cinco et al., 2016), and recently the southern Philippines has seen a dramatic increase in the rate of landfalls from December-February (Basconcillo
and Moon, 2021). Any changes in the frequency, intensities and other characteristics of TCs due to anthropogenic climate change are therefore of extreme importance for the Philippines.

TCs occur in several ocean basins around the world, including the Western North Pacific (WNP), in which the Philippines lies, and the North Atlantic (NA), among others. The influence of anthropogenic climate change on TCs varies by basin, as does the level of scientific evidence on these changes (Masson-Delmotte et al., 2021). On a global scale, recent decades have seen
an increase in more intense TCs (category 3-5 on the Saffir-Simpson scale), but no change in the overall number of TCs. Recent studies on specific TCs as well as physical understanding suggest that extreme rainfall from TCs is increasing (Masson-Delmotte et al., 2021). This is explained partly by the Clausius-Clapeyron relationship, which states that warmer air holds more moisture at a rate of ~7% / °C. In the future, the proportion of the most intense TCs (categories 4-5) is projected to increase with further warming, as well as the average and maximum precipitation rates from these storms (Seneviratne et al.,
75    2021).

Basin-specific changes are less certain, more variable and extend to other properties of TCs. For instance, in the NA, TCs making landfall over the contiguous US have slower translation speeds (Kossin, 2018) and are more frequently stalling or meandering, leading to more intense impacts as extreme conditions are sustained for longer periods over a given location (Seneviratne et al., 2021). In addition, the NA hurricanes are increasing in both intensification rate and maximum intensity,
which is unlikely to be explained by natural variability (Bhatia et al., 2019; Murakami et al., 2020). Evidence is scarcer in the WNP, due in part to the many ways in which anthropogenic climate change may influence TCs and their associated meteorological hazards. Research suggests that there is no significant trend in the rate of landfalling TCs in the Philippines





(Cinco et al., 2016), and that TCs are moving more slowly (potentially leading to enhanced extreme conditions), and systematically tracking further north (poleward) (Kossin et al., 2016; Yamaguchi and Maeda, 2020).

Ultimately 'natural' hazards such as TCs only become disasters due to the exposure and vulnerability of people and the built environment to these hazards. The Philippines has a rapidly growing population of nearly 120 million people in 2024, which is increasingly urbanised. The growth of urban populations and relatively high rates of poverty has led to the growth of informal settlements that are unable to withstand extreme weather conditions, as well as other sources of pervasive housing-based vulnerability (Healey et al., 2022). The landfall of Typhoon Haiyan (known locally as 'Yolanda') in 2013 also highlighted this

(Santos et al., 2015). Haiyan devastated coastal areas in the eastern Visayas, creating cascading impacts by displacing over 4 million people, destroying or damaging over a million homes, and damaging infrastructure that in turn disrupted key services (Lagmay et al., 2015; USAID, 2025; Yi et al., 2015). A lack of insurance penetration among farmers (due to high premiums) also compounded severe damage to the valuable coconut farming sector (Seriño et al., 2021).

Given this high exposure and vulnerability, the Philippines is acutely susceptible to influences of anthropogenic climate change

upon TC hazards. It is therefore crucial to isolate potential changes up to date and projected future changes, to enable estimates of the costs of anthropogenic climate change for the Philippines to date and to assess risks going forwards. As set out above, current evidence for overall TC trends in the WNP basin remains limited. However, the changing hazards and risks from destructive storms like those that have already occurred can be explored forensically through the science of extreme event attribution.

**1.3 Extreme event attribution of tropical cyclones**

Extreme event attribution (EEA) is a field of science that allows the assessment of whether and to what degree anthropogenic climate change influenced the likelihood and/or intensity of an extreme weather event, or some aspects of the event, or a class of similar events at or above a certain rarity/intensity. This may find that anthropogenic climate change made an event less likely/intense or doesn't play a significant role. For instance, the extreme cold surge in Northeast China in April 2020 was

made about 80% less likely by anthropogenic climate change (Yu et al., 2022). Studies also find that anthropogenic climate change has not played a significant role in an event or do not have sufficient evidence to conclude firmly. For example, strong windstorms in western Europe in January 2018 were not significantly affected by anthropogenic climate change (Vautard et al., 2019). Finally, many studies find that anthropogenic climate change has increased the likelihood and/or intensity of an event, such as the record-breaking North American Northwest heatwave in 2021 that led to hundreds of deaths in western

Canada, which would have been effectively impossible without anthropogenic climate change (Philip et al., 2022). Specifically, EEA enables an assessment of the risk of an event occurrence (or a class of events) and the factors driving changes in it that is based on both lived experience and an event that is known to be plausible in the present.





Several attribution studies exist for TCs around the world, but the number of events studied varies significantly by the region and ocean basin in which they occurred. Hurricanes in the NA basin are the most frequently studied TC events to date, with
studies covering 8 recent events with combined damages of more than US$600 bn (at the time of occurrence) in the US and Caribbean. Almost across all the studies, rainfall from these events were amplified by anthropogenic climate change: Katrina in 2005 by 4%, Irma in 2017 by 6%, Maria in 2017 by 9% (Patricola and Wehner, 2018), Florence in 2018 by 5% (Reed et al., 2020), Dorian in 2019 by 5-18% (Reed et al., 2021), Ian in 2022 by 18% (Reed and Wehner, 2023) and Harvey in 2017 by 7-38% (Emanuel, 2017; Van Oldenborgh et al., 2017; Risser and Wehner, 2017; Wang et al., 2018). Meanwhile, during
Hurricane Sandy in 2012, the effect of sea level rise due to anthropogenic climate change created a higher storm surge, directly leading to an additional US$8.1 bn in damages (Strauss et al., 2021).

The NA basin has attracted many studies for a range of reasons: the large number of events making landfall and causing high economic damages, the relative abundance and reliability of data records for historical events, the funding available to experts to study TCs in the region, the consistency in documenting and recording of impacts from TCs. As set out above, events in the
NA, largely striking the eastern seaboard of the US and the Caribbean, have consistently caused the largest monetary damages in the world. The impacts of TCs in economic terms are partly a function of the exposed property. Thus, the absolute economic impacts tend to be higher for more developed nations with more assets at risk.

However, for people on the ground, the severity of an event is not measured by the overall economic damages but by impacts on lives, health (physical and mental), livelihoods and personal property (especially where property is uninsured) and access
to crucial services. Some of the most devastating TCs in terms of loss of life, injury, displacement, disruption and other impacts have occurred in other basins. For instance, in 2008, Cyclone Nargis formed in the Bay of Bengal and made landfall in Myanmar, leading to widespread devastation and the deaths of at least 138000 people (Howe, 2019). In 2013 in the Philippines, in addition to the impacts described above, Typhoon Haiyan resulted in 6000 deaths and 30000 injuries (ReliefWeb, 2025; Lagmay et al., 2015).

Reflecting the severe damage and risk to people's lives and livelihoods posed by events in other basins, recent years have seen the extension of attribution studies to events in these regions. This includes the WNP basin in which the Philippines lies and therefore forms a crucial first layer of evidence and foundational point for further studies. To date, attribution studies for this basin include Typhoons Hagabis (2019), Morakot (2009), Bopha (2012), Mangkhut (2018) and the previously described Typhoon Haiyan (2013), as well as a series of events striking Vietnam in 2020, and the anomalously active cyclone season of
2015 (figure 2).



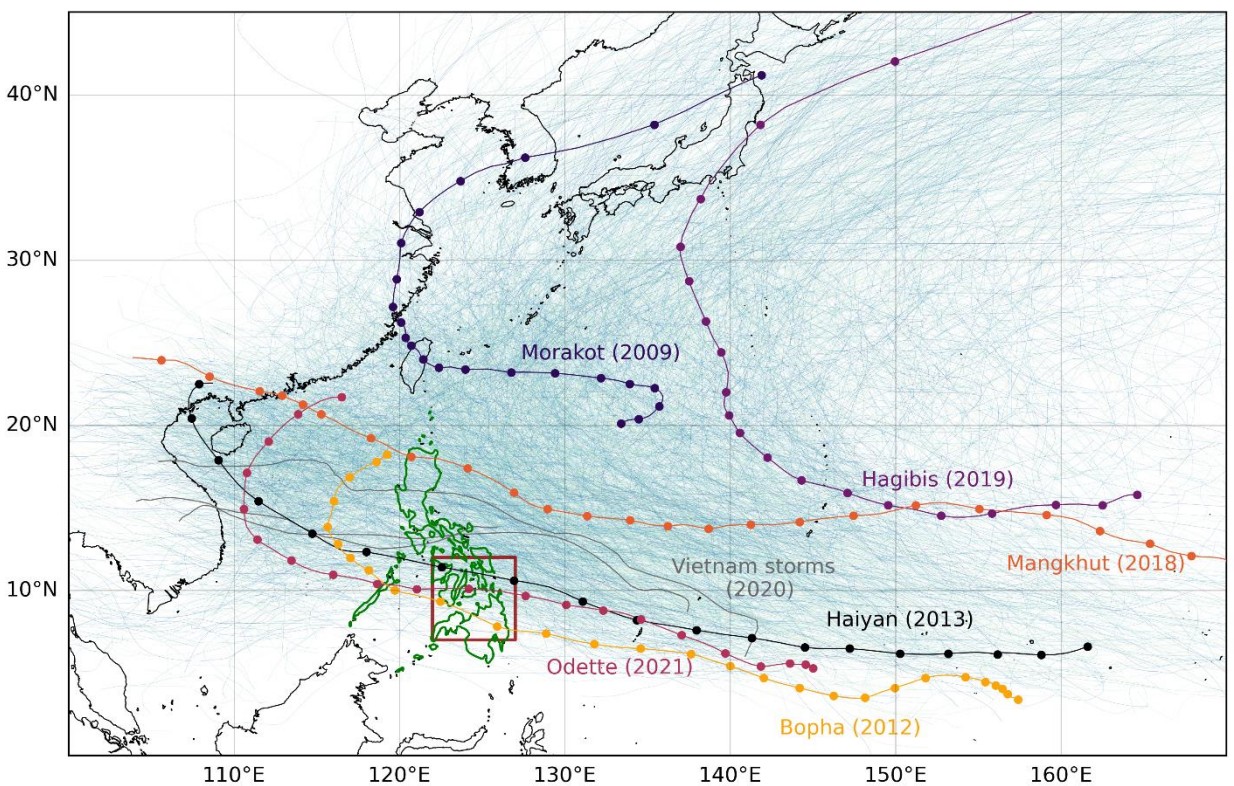

*Figure 2: Storm tracks over the observed historical period from IBTrACS*

*(https://www.ncei.noaa.gov/products/international-best-track-archive). The highlighted storm tracks are for those with*

*attribution studies, including Odette (studied here), Haiyan, Morakot, Hagibis, and a series of storms striking Vietnam in*

*October 2020. The Philippines is outlined in green and the study region for the attribution analysis in Sect. 3 is highlighted*

*in dark red.*

The picture for TCs in the WNP is less consistent than in the NA, though most studies show that anthropogenic climate change has amplified the likelihood and/or intensity of TCs. For instance, in the 2015 TC season, the accumulated cyclone energy was linked to anthropogenically-amplified sea surface temperatures (SSTs) in the eastern and central Pacific (Zhang et al., 2016).

The extreme rainfall associated with Typhoon Hagibis, which affected the Tokyo region in 2019, has been tested for an anthropogenic influence using two complimentary methods. A probabilistic approach found that 15-150% more likely to occur because of anthropogenic climate change, equating to a fractional attributable risk (FAR) of 0.4 and thus approximately US$4 billion in climate change-related damages (Li and Otto, 2022). Meanwhile, a storyline approach - a method used in event attribution to understand the causal chain leading to an extreme event focusing on the physical processes that contributed to

the event - found that anthropogenic climate change amplified the rainfall by around 11% (Kawase et al., 2021). Typhoon



Morakot was another major event in the basin, which made landfall in Taiwan in August 2009, bringing extreme rainfall. By simulating the same storm with and without late century (1985-2005) warming, researchers found that anthropgenic warming in this period directly led to an increase of approximately 3.5% in Morakot's total rainfall (Wang et al., 2019). In 2020, in central Vietnam, while no single event caused major impacts, a sequence of typhoons typhoon-induced extreme rainfall events

led to severe impacts. However, a probabilistic approach found that there was no detectable anthropogenic influence on multi-week rainfall accumulations in central Vietnam (Luu et al., 2021).

In the Philippines, studies have largely considered changes in intensity. Typhoons Bopha, Mangkhut and Haiyan were studied using a 'pseudo-global warming' approach, which showed that the maximum winds from these storms were increased by 10 m/s, 2 m/s and 2 m/s, respectively, relative to preindustrial times (Delfino et al., 2023). The influence of anthropogenic climate

change (ACC) on the intensity and storm surge from Typhoon Haiyan is analysed by multiple other studies (Delfino et al., 2023; Nakamura et al., 2016; Takayabu et al., 2015; Wehner et al., 2019), finding conflicting influences due to the relative contribution of different drivers, as well as storm- and model- dependency. More recent analysis using a synthetic tropical cyclone model, which does not rely on explicit simulation by climate models, found that a typhoon with a landfall maximum wind speed like Haiyan was very rare at the time it occurred, with a return period of approximately 850 years (Sparks and

Toumi, 2025). Further, this analysis showed that it would have been nearly impossible in a world without anthropogenic climate change, with a change in likelihood of such events of a factor of around 45, relative to preindustrial conditions. Equivalently, events of such probability have become approximately 3.5 m/s more intense due to ACC.

## 1.4 This study

In this study, a range of attribution methods are employed to disentangle the influence of anthropogenic climate change on

Super Typhoon Odette (henceforth Odette). First, in Sect. 2 the use of circulation analogues is explored. This method employs the observed circulation patterns during Odette and studies comparable events, i.e., with similar sea level pressure anomaly patterns, in high resolution global climate models, under climate conditions of past, present and future. This method serves two purposes of assessing: a) whether the current generation of high resolution climate models (with resolution on par of those analysed in the method of Sect. 3, as detailed below), can capture the circulation patterns leading to events comparable to Odette; and b) whether there are changes in the anomaly's strength and/or spatial extent, hinting at possible changes in cyclone

behaviour, under different climate conditions with different levels of anthropogenic forcing. This method provides a relatively quantitative assessment of the ability of the current high-resolution models, and if/how the extreme rainfall and strong winds change given similar circulation analogues.

Second, in Sect. 3, a probabilistic event attribution for extreme rainfall as observed during Odette is conducted. This comprises

a statistical analysis of changing extremes using the observed magnitude of Odette as a basis. It employs both gridded



observational products and regional climate models to assess how extremes are changing due to changes in global mean surface temperature (GMST).

Third, in Sect. 4, the changing likelihood of landfalling category 5 tropical cyclones is modelled using a synthetic tropical cyclone model, based on IBrACS observations. This is the Imperial College Storm Model – IRIS – described in the IRIS model
description paper (Sparks and Toumi, 2024), Haiyan study (Sparks and Toumi, 2025) and further in Sect. 4.

Taken together, this study combines several complementary approaches to attributing the meteorological hazards associated with Odette: a conditional attribution focusing on events of similar circulation patterns, a probabilistic analysis of changing likelihoods of precipitation extremes, and a probabilistic analysis grounded in historical events and physical understanding of strong wind extremes.

Finally, the implications of this work for the attribution of impacts from Odette are discussed. This involves synthesising the multi-method analysis of the meteorological hazards, a discussion of the compounding nature of multiple hazards, and the factors leading from these hazards to impacts on the ground. As part of this, avenues for further work are also discussed.

## 2 Circulation analogues

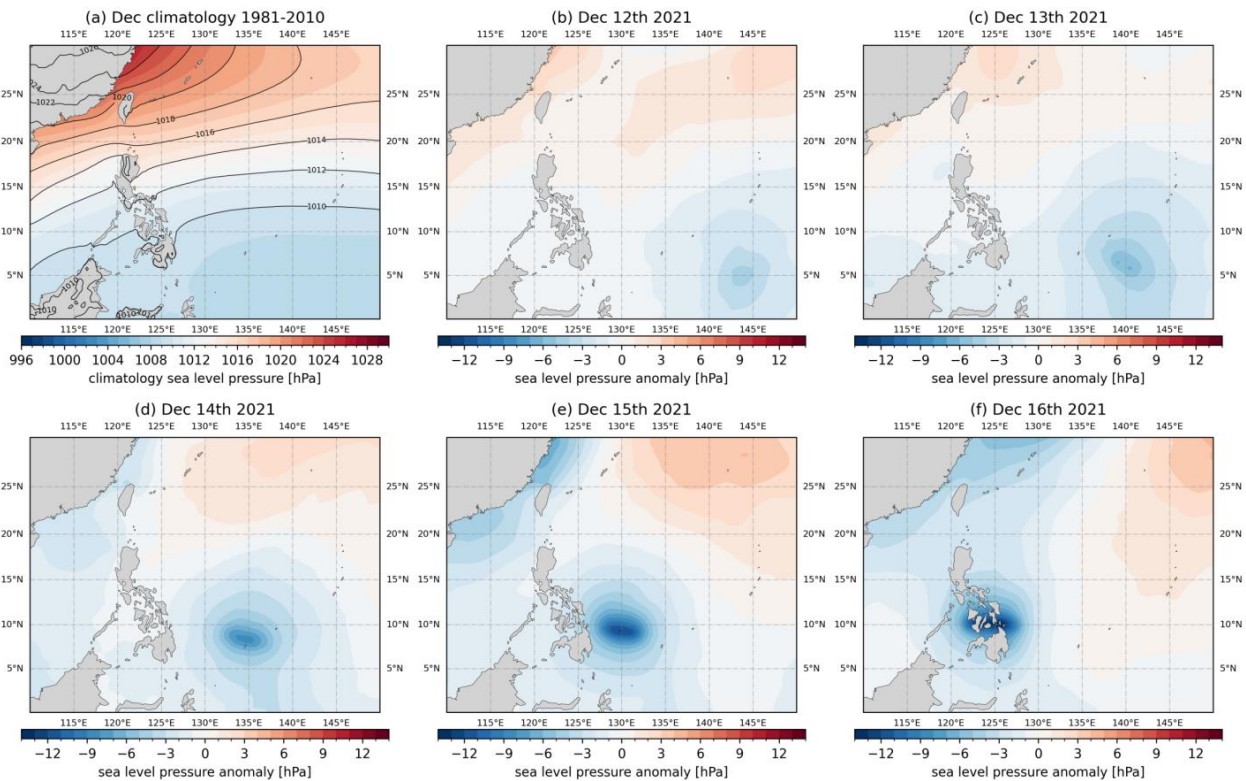



*Figure 3: The climatological mean over 1981-2010 Dec sea level pressure (SLP) fields (in hPa) in the upper left panel (a), and the evolution of anomalous sea level pressure (w.r.t. climatological mean shown in panel a) associated with the formation and passing through of Odette from Dec12th to Dec16th of 2021 in panels (b) through (f).*

Circulation analogues are other occurrences of similar atmospheric conditions, often shown using atmospheric pressure levels, to that in which the event in question occurred. Comparing the meteorological conditions under similar circulation regimes but at different times (or warming levels) helps to control for any changes in such dynamics by fixing the dynamical conditions and thus testing only the direct 'thermodynamic' contribution of warming to an event. Here, testing for the existence of circulation analogues within models also helps to evaluate whether models can capture such conditions.

Figure 3 shows sea level pressure evolution associated with Odette. The early development of Odette (panel b) is visible as a localized area of negative sea-level pressure anomalies (~-4 hPa) southeast of the Philippines on the 12[th] of Dec. The negative anomalies intensify (~-6 hPa) on the 13[th], and the system becomes more organized. By the 14[th], the low-pressure anomaly has deepened significantly (~-8 to -10 hPa), with a well-defined structure, and by the 15th, the system reaches its peak intensity with central pressure anomalies exceeding -12 hPa. This extreme anomaly signifies the fully developed typhoon with tightly packed isobars, consistent with the heavy rainfall and strong winds associated with it. On the 16[th], Odette makes landfall in the Philippines, and the typhoon's core pressure remains significantly anomalous (~-10 to -12 hPa), The persistence of strong negative anomalies over the region reflects the typhoon's ongoing impact. Using Figure 3 as a guide, this section focuses on the sea level pressure (hPa) anomaly patterns associated with Typhoon Odette, particularly Dec 16th - the day of landfall during which most disastrous impacts occurred (as detailed in the introduction). The circulation analogue method is based on looking for circulation conditions similar to those that caused the damaging heavy rainfall (Sect. 3) and strong winds (Sect. 4), under both current climate and past climate conditions.

The analogue is defined as the spatial pattern of sea level pressure anomaly over the domain of interest (5-20 °N, 115-130 °E), given this is the key region of the low-pressure anomaly on the 16[th]. We have also tested the sensitivity of the results to expanding the size of the domain to (5-20 °N, 115-135 °E) (results shown in the Appendix, Figure A1), and the results do not present substantial differences. The analogues search is only performed on sea level pressure anomaly patterns of Dec 16th, given that Odette made landfall in the Philippines on that day, and to be consistent with sections 3 and 4.

In this section, simulations from the sixth phase of the Coupled Model Intercomparison Project (CMIP6) generation experiments, the High-Resolution Model Intercomparison Project (HighResMIP) SST-forced model ensemble (Haarsma et al. 2016) is selected for analysis, with simulations spanning from 1950 to 2050. HighResMIP provides multi-model and multi-resolution simulations (Haarsma et al. 2016). Under the CMIP6 HighResMIP protocol, the European Union Horizon 2020 project, PRIMAVERA (PRocess-based climate sIMulation: AdVances in high resolution modelling and European climate



Risk Assessment), has provided global atmospheric general circulation model (AGCM) simulations at CMIP6-standard resolution (~100 km) and higher resolutions (~25 km). These simulations enable scientists to analyse TCs and assess the
reliability of changes in TC rainfall across a variety of numerical models and spatial scales (Roberts et al. 2020). The SST and sea ice forcings for the period 1950-2014 are obtained from the 0.25° x 0.25° Hadley Centre Global Sea Ice and Sea Surface Temperature dataset that are area-weighted regridded to match the climate model resolution (see Table 1). For the 'future' time period (2015-2050), SST/sea-ice data are derived from RCP8.5 (CMIP5) data, and combined with greenhouse gas forcings from SSP5-8.5 (CMIP6) simulations (see Sect. 3.3 of Haarsma et al. 2016 for further details). The models used in this
analysis are detailed in Table 1 below.

| Model | Resolution | Institute |
|---|---|---|
| CMCC-CM2-VHR4 | ~25 km | Fondazione Centro Euro-Mediterraneo sui Cambiamenti Climatici |
| CMCC-CM2-HR4 | ~100 km | Fondazione Centro Euro-Mediterraneo sui Cambiamenti Climatici |
| CNRM-CM6-1-HR | ~50 km | Centre National de Recherches Meteorologiques |
| CNRM-CM6-1 | ~100 km | CNRM-CERFACS |
| EC-Earth3P-HR | ~40 km | EC-Earth-Consortium |
| EC-Earth3P | ~80 km | EC-Earth-Consortium |
| HadGEM3-GC31-HM | ~25 km | UK Met Office, Hadley Centre |
| HadGEM3-GC31-MM | ~60 km | UK Met Office, Hadley Centre |
| HadGEM3-GC31-LM | ~135km | UK Met Office, Hadley Centre |
| MPI-ESM1-2-XR | ~60 km | Max Planck Institute for Meteorology |



| MPI-ESM1-2-HR | ~100 km | Max Planck Institute for Meteorology |
|---|---|---|

*Table 1: List of HighResMIP models used in the study.*

We use observation-based reanalysis ERA5 to characterise the anomalously low sea level pressure patterns (Figure 3) and use HighResMIP models to investigate the possible changes of events similar to Odette. We search for if and how frequently these

circulation patterns occur in different time periods. We consider a "past" 20-year time slice of 1950-1970, a "present" 20-year time slice of 2001–2021, and a "future" 20-year time slice of 2030-2050. The past represents a world with a weaker anthropogenic influence on climate than the present, the present refers to the current climate conditions, and the future represents a future world with further warming and enhanced anthropogenic climate change compared to the present.

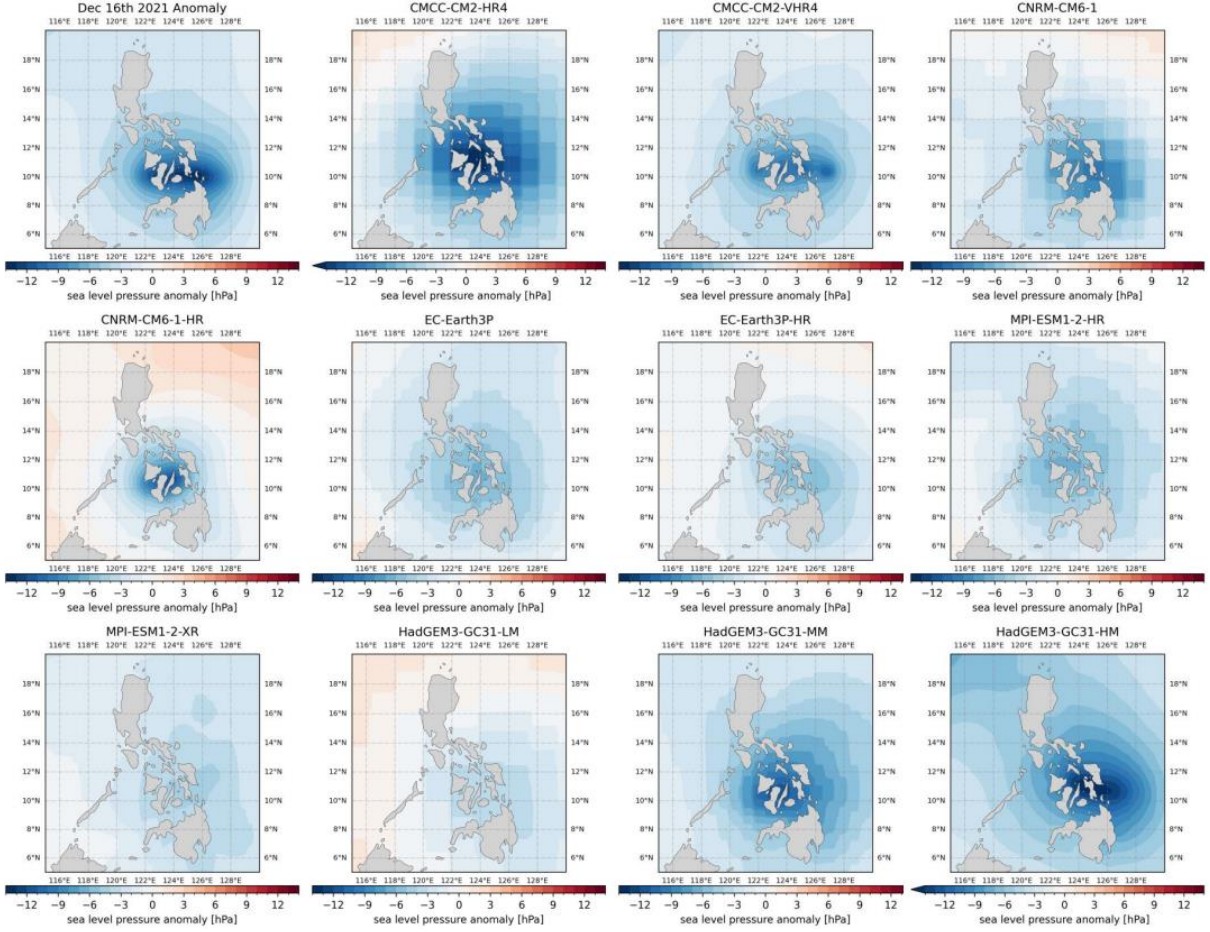

*Figure 4: For each model, the sea level pressure anomaly patterns over (5-20 °N, 115-130 °E), with the highest spatial correlation to Dec 16th 2021 during Odette landfall is shown.*



Figure 4 displays the sea-level pressure (SLP) anomaly patterns from various models, each showing the pattern with the highest spatial correlation to the observed anomalies during Typhoon Odette's landfall on December 16th, 2021. The panels illustrate how different models simulate the SLP anomalies in the region (5°–20°N, 115°–130°E), providing insight into the models' ability to capture the cyclone's intensity and structure. The observed SLP anomalies highlight a strong negative anomaly (~-12 hPa) centred over the Philippines, marking the typhoon's core. CMCC-CM2-HR4 and CMCC-CM2-VHR4 capture a negative anomaly over the Philippines, but the intensity and spatial extent are less pronounced compared to observations. The higher-resolution version CNRM-CM6-1-HR better represents the observed central low, while the lower-resolution version CNRM-CM6-1 shows a weaker anomaly. Both EC-Earth3P and EC-Earth3P-HR show negative anomalies, but the higher-resolution version of the model (HR) offers a more concentrated anomaly, aligning more closely with the observed typhoon core. The MPI models simulate a similar anomaly pattern but much weaker compared to the observed. Among all models, the high-resolution HadGEM3-GC31-HM best reproduces the observed anomaly intensity and spatial pattern, closely matching the central low (~-12 hPa). Higher-resolution models generally provide more accurate simulations of the SLP anomalies, capturing the intensity and compact structure of Typhoon Odette; whereas models with coarser resolutions tend to underestimate the strength of the negative anomaly and spread it over a larger area. These results further demonstrate the importance of high-resolution modelling in accurately representing extreme tropical cyclone events and their associated sea level pressure patterns. Given these results, the following analysis only use the high-resolution version (CMCC-CM2-VHR4, CNRM-CM6-1-HR, EC-Earth3P-HR, and HadGEM3-GC31-HM, all of which with <=50km horizontal resolution as detailed in Table 1), while excluding the MPI models given the weak comparison with the observed.





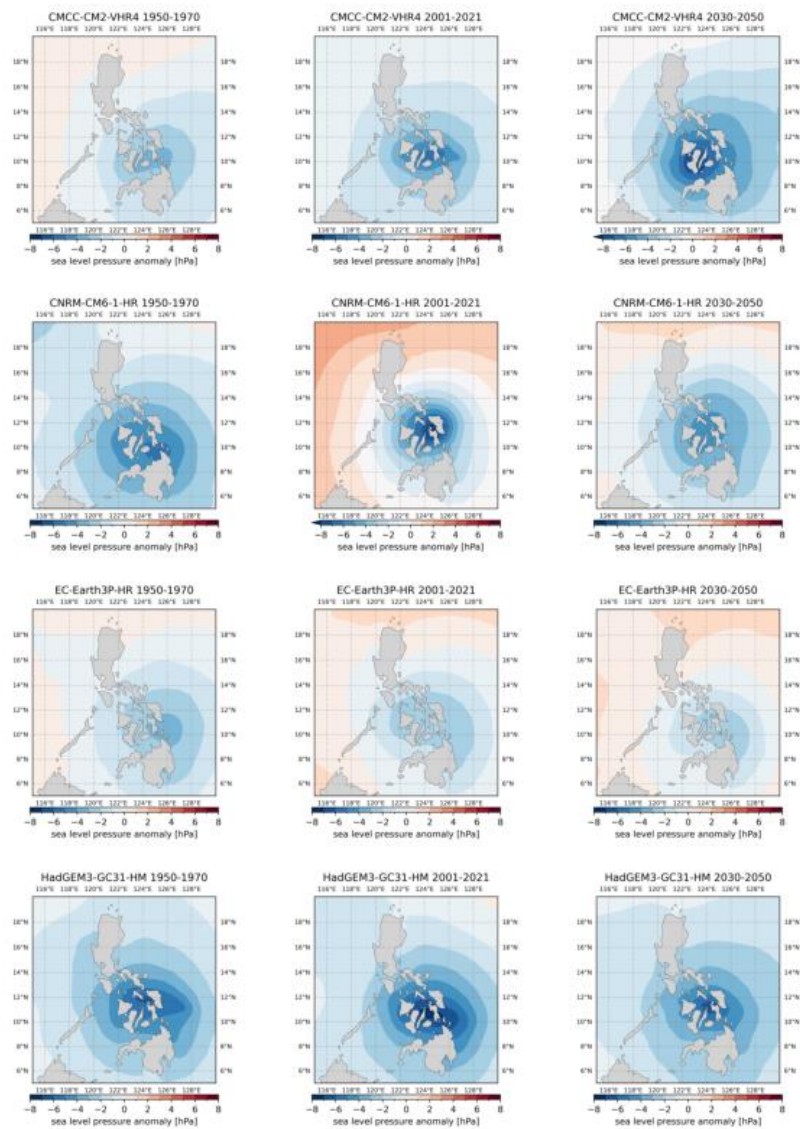

*Figure 5: Composite maps of the top 5 ranking analogues for each of the selected model (rows) and each period past (left), present (middle) and future (right).*


Figure 5 presents the composite maps of the top 5 ranking analogous for each of the selected models across the past, present and future. Across different periods, the models generally maintain a consistent pattern, suggesting a robust ability to simulate this kind of patterns across different time slices. However, the 2030-2050 projections sometimes show a shift in the anomaly's strength and location, suggesting possible future changes in cyclone behaviour. For example, CMCC-CM2-VHR4 predicts

slightly stronger and more expansive anomalies, which could indicate more intense and larger spatial extent of cyclones,





whereas CNRM-CM6-1-HR predicts a more expansive but weaker anomaly. Both EC-Earth3P-HR and HadGEM-GC31-HM predicts a less expansive and slightly weaker anomaly. The differences among models highlight the uncertainty in projections and the importance of using multiple models to capture the range of possible future outcomes.

Despite the overall consistency in capturing the anomaly, the variability between models and across time periods indicates uncertainty in both the models' ability to simulate this kind of event and in future projections. The range of different model behaviours illustrates the challenges in predicting the exact impact of anthropogenic climate change on tropical cyclone behaviour in this region with this set of models. In summary, the comparison of the observed sea-level pressure anomaly with the model ensemble suggests that this set of climate models can simulate the general features of such extreme cyclonic events,

though with some variation in intensity and location. The future projections indicate that these anomalies may become more/less pronounced or shift in other characteristics such as spatial extent, pointing to the need for continued research into the impacts of anthropogenic climate change on tropical cyclones in the Philippines. These results further highlight the importance of using a multi-model ensemble to capture the range of potential future outcomes, as has been done in this study.




*Figure 6: Daily precipitation (left), daily maximum wind speed (middle), daily mean wind speed (right) corresponding to the top 5 ranking analogues shown in Fig. 5, for past (dark blue), present (green) and future (light blue).*





Given the circulation analogues shown in Figure 5, Figure 6 presents the corresponding shift of daily precipitation, daily
maximum wind speed, and daily mean wind speed for the selected models across the three different time period. Given the
data sample size is only 5, the standard deviation in this figure is only for illustrative purposes. For daily precipitation, across
all models, there is a general increase in daily precipitation from 1950-1970 to 2001-2021. This suggests a trend toward wetter
conditions given similar circulation conditions-consistent with thermodynamic understanding from Clausius–Clapeyron, with
some models showing more pronounced increases than others (e.g., CNRM-CM6-1-HR). EC-Earth3P-HR is the only model
showing a slight decrease in daily precipitation from 1950-1970 to 2001-2021. There is a moderate increase in daily maximum
wind speeds in most models. This indicates a rise in the intensity of extreme winds, which could be associated with stronger
storms due to anthropogenic climate change. Similarly, all models show a slight increase in daily mean wind speeds, suggesting
that winds have become somewhat stronger on average over the past few decades. These results suggest that there is relatively
high confidence that anthropogenic climate change has led to increase in precipitation and winds under similar circulation
conditions over the past few decades.

Comparing 2001-2021 to 2030-2050, the picture is less clear. The changes in daily precipitation across models are mixed, with
some models (e.g., CNRM-CM6-1-HR, HadGEM3-GC31-HM) predicting a slight increase, while others (e.g., CMCC-CM2-
VHR4, EC-Earth3P-HR) predict a slight decrease. Most models predict a slight decrease in daily maximum wind speeds,
indicating that extreme winds may become less intense over the coming decades. The changes in daily mean wind speeds are
mixed: some models (e.g., HadGEM3-GC31-HM, CNRM-CM6-1-HR) project a slight increase, while others (e.g., CMCC-
CM2-VHR4, EC-Earth3P-HR) project a slight decrease. In summary, the future projections for daily precipitation and wind
speeds indicate a complex picture for the period 2030-2050. While some models suggest increasing precipitation and wind
speeds, others predict a slight reduction. The overall trends suggest a high degree of variability.


The changing properties of extremes under similar synoptic conditions is not the primary conclusion of this section. Rather,
it's that the existence of analogues in the current generation of climate models indicates that they are applicable for studying
certain elements of TCs (though remain inapplicable for others). For instance, the phenomena known to generate the most
extreme winds and localised rainfall from TCs occur at scales of ~1-10 km. The damage from winds tends to result from the
most intense localised magnitudes, making this limitation prohibitive for analysing extreme winds. However, rainfall over a
larger region, over tens to hundreds of kilometres, is both captured well and is still relevant to many of the impacts of such
events. Specifically, in the following section similar climate models to those studied here are used to analyse changes in
extreme rainfall at the scale of several hundred kilometres. This section therefore provides additional confidence that the
probabilistic event attribution modelling conducted in this paper is relevant to this study of the influence of anthropogenic
climate change on Typhoon Odette.





## 3. Attribution of extreme rainfall using probabilistic event attribution method

### 3.1 Event definition

The first step in a probabilistic event attribution (PEA) study is to define the event in question. The central aim is to link the
event definition as closely as possible to the impacts of the event on the ground. In this case, that means studying both high
winds and extreme rainfall, as impacts arose due to a combination of these meteorological hazards. Furthermore, the impacts
were greatest in parts of the Visayas around the south-central Philippines. While the impacts were only experienced on land,
the storm surge arose due to high winds over the ocean, and the land masses in this region are extremely complex on small
scales. Most gridded observational products and regional climate models are too coarse to capture these coastal terrain features
accurately. As a result, the region selected is a box bounded by 7 - 12°N, 122 - 127 °E (including both land and ocean areas
within this), which captured the passage of the event on December 16th of 2021, in extreme precipitation (and maximum wind
speeds - shown in figures 1 & 2).

Finally, the event definition chosen here is limited to single day maxima in accordance with observations of the event (figure
1), and over two time-windows of the year: December only, and the wider June-December period. The former enables an
assessment of changes in TCs at the end of the season when such extreme TCs are less common. The latter period captures the
active TC season for most of the Philippines, ensuring that the most occurrences of high precipitation/strong winds are
associated with TCs, while also including the month of December in which the event occurred. Using multiple timescales helps
to test the sensitivity of results to the event definition selection and provides additional insight into the changing risk of the
meteorological hazards. Finally, across the four observational datasets used (Sect. 3.2.1), TC events account for 70-80% of all
extreme rainfall events in the studied period. Confidence is therefore high that any detected changes are mostly due to changes
in TCs. Furthermore, ultimately the impact on the ground is irrespective of the source of the rainfall; Odette is part of a class
of high rainfall events leading to impacts in the Philippines, some of which have different meteorological origins, but all of
which may lead to comparably high damage.

### 3.2 Data and methods

### 3.2.1 Observational data

In this section, we use a range of gridded observational and reanalysis products: the ERA5 reanalysis product (Hersbach et al.,
2020), from which we use daily precipitation and maximum wind speed data; the Multi-Source Weighted-Ensemble
Precipitation (MSWEP) v2.8 dataset (updated from Beck et al., 2019) at 0.1° spatial resolution, available from 1979 to ~3
hours from real-time; the "Climate Hazards Group InfraRed Precipitation with Station data" (CHIRPS; (Funk et al., 2015), at
0.05° resolution, from 1981 to 31st Jan 2024; the Global Precipitation Climatology Centre (GPCC) Full Data Daily Product





Version 2022 of daily global land-surface precipitation totals at 1.0° (GPCC Full Data Daily Version 2022); finally, as a measure of anthropogenic climate change we use the (low-pass filtered) global mean surface temperature (GMST) from the Goddard Institute for Space Science (GISS) surface temperature analysis (GISTEMP, Hansen et al., 2010 and Lenssen et al., 2019).


### 3.2.2 Model and experiment descriptions

In this section, we use 2 multi-model ensembles from regional climate modelling experiments with model domains covering the Philippines. All of these simulations are composed of historical simulations up to 2005 and extended to the year 2100 using the RCP8.5 scenario under CMIP5 forcings.


1. Coordinated Regional Climate Downscaling Experiment-Southeast Asia (CORDEX-SEA) (0.22° resolution, SEA-22) multi-model ensemble (Supari et al., 2020), consisting of 8 simulations resulting from pairings of 4 Global Climate Models (GCMs) and 3 Regional Climate Models (RCMs).


2. Coordinated Regional Climate Downscaling Experiment-East Asia (CORDEX-EAS) (0.22° resolution, EAS-22) multi-model ensemble (Kim et al., 2021), consisting of 6 simulations resulting from pairings of 3 Global Climate Models (GCMs) and 2 Regional Climate Models (RCMs).

### 3.2.3 PEA statistical methods

We analyse time series from a region averaged over the southern Philippines (a box bounded by 7 - 12 °N, 122 - 127 °E, see figures 1 & 2), of rainfall during December and June-December, where long records of observed data are available. Methods for observational and model analysis and for model evaluation and synthesis are used according to the World Weather Attribution Protocol, described in Philip et al. (2020), with supporting details found in van Oldenborgh et al., (2021) and Ciavarella et al. (2021).

The analysis steps include: (i) trend calculation from observations; (ii) model validation; (iii) multi-method multi-model attribution and (iv) synthesis of the probabilistic attribution statement. We calculate the return periods, Probability Ratio (PR; the factor-change in the event's probability) and change in intensity of the event under study to compare the climate of now and the climate of the past, defined respectively by the GMST values of now and of the preindustrial past. Using climate models only, we also use the same method to compare the studied event in the present and a hypothetical future climate at a 390 GMST value of 2 °C, i.e., approximately 0.8 °C above present levels. To statistically model the event under study, we use a generalised extreme value distribution that scales with GMST. As a final step, results from observations and models that pass the validation tests are synthesised into a single attribution statement.



### 3.3 Observational analysis

**3.3.1 December**

Table 2 (and figure B1) show the trends in December maxima of 1-day rainfall over the study region for four gridded observational and reanalysis datasets. All datasets show a strong increase in the likelihood of events with 1-day rainfall at least as extreme as Odette with global warming, with estimates of the PR ranging from 3.8 in ERA5 to over 1000 in CHIRPS and MSWEP, with MSWEP being a statistically significant estimate. Combining all estimates, a return period of 20 years is used

to characterise the event in model analysis.

| Dataset | Observed event | | Trend due to GMST | |
|---|---|---|---|---|
| | Magnitude (mm) | Return period (95% C.I.) | Probability Ratio (95% C.I.) | Change in magnitude (%) (95% C.I.) |
| ERA5 | 83.26 | 30.61 (9.9 – 12625) | 3.8 (0.14 – 469.64) | 31.6 (-20.49 – 133.52) |
| CHIRPS | 76.48 | 11.64 (4.5 – 206.82) | 1596.9 (0.84 – inf) | 63.41 (-5.27 – 158.98) |
| MSWEP | 78.94 | 20.06 (5.55 – 757.37) | 1680.9 (1.64 – inf) | 97.35 (4.67 – 331.41) |
| GPCC | 76.98 | 19.58 (5.76 – 522.38) | 9.28 (0.34 – inf) | 43.07 (-18.5 – 166.09) |

*Table 2: Observed event magnitude and return period of the events in the present climate (2021), and the estimated probability ratio and change in magnitude of such an event due to global warming of 1.2 °C, across four datasets. Event magnitudes are reasonably consistent across datasets. Increasing (decreasing) trends due to GMST are highlighted in blue (orange).*


**3.3.2 June-December**

Figure 7 and table 3 show the trends in June-December maxima of 1-day rainfall over the study region for four gridded observational and reanalysis datasets. Datasets show mixed trends in the likelihood of events at least as extreme as Odette with global warming. Estimates of the probability ratio range from a decrease of a factor of 2 in MSWEP up to an increase of

infinity in CHIRPS, suggesting the event would have been statistically impossible without global warming. None of the



estimates are statistically significant. Combining all estimates, again a return period of 20 years is used to characterise the event in model analysis.

| Dataset | Observed event | | Trend due to GMST | |
|---|---|---|---|---|
| | Magnitude (mm) | Return period (95% C.I.) | Probability Ratio (95% C.I.) | Change in magnitude (%) (95% C.I.) |
| ERA5 | 83.26 | 24.3 (9.9 – 135.0) | 1.56 (0.28 – 14.96) | 8.32 (-18.53 – 46.64) |
| CHIRPS | 76.48 | 31.4 (6.7 – inf) | inf (0.05-inf) | 22.86 (-29.12 – 77.74) |
| MSWEP | 78.94 | 21.1 (7.4 – 174.6) | 0.46 (0.04 – 267.26) | -13.61 (-46.68 – 50.65) |
| GPCC | 76.98 | 13.8 (4.0 – 74.4) | 0.6 (0.06 - inf) | -8.86 (-49.28 – 55.59) |

*Table 3: Observed event magnitude, return periods in past (-1.2 C) and present climates (2021), and the estimated probability ratio of such an event, across four datasets. Event magnitudes are reasonably consistent across datasets.*






*Figure 7: Statistical fits and trends in the four observational datasets, ERA5, CHIRPS, MSWEP and GPCC for June-December. a) Shows the GEV fit to the ERA5 data at two levels of the covariate GMST: in 2021 (red line) and in a 1.2 °C cooler climate (blue line). The purple line shows the magnitude of Odette. b) Shows the estimated trend, with the event associated with Odette highlighted in purple. c) and d), e) and f), g) and h): As in (a) and (b) for CHIRPS, MSWEP and GPCC data.*

From observations alone, there is a consistent trend showing that global warming has increased the likelihood and intensity of extreme rainfall events like Typhoon Odette in December alone. However, it remains unclear whether this effect extends to the wider TC season of June-December due to disagreement between datasets. To further elucidate the link between anthropogenic climate change and Typhoon Odette, the results from observations for each event definition will be synthesised with the respective results from climate models. This allows an overall estimate of the influence of anthropogenic climate change that is not as heavily dependent on individual data sources.

## 3.4 Model evaluation

The climate models are evaluated against the observations in their ability to capture:

1. Seasonal cycles: Qualitative comparison of seasonal cycles based on model outputs against observations-based cycles, discarding models that exhibit multi-modality and/or ill-defined peaks in their seasonal cycles.

2. Spatial patterns: Similarly, models that do not match the observations in terms of the large-scale annual mean precipitation patterns are excluded, with a focus on the E-W gradient of rainfall across the Philippines.

3. Parameters of the fitted GEV models: Models are discarded if the model and observation parameters ranges do not overlap.

The models are labelled as 'good', 'reasonable', or 'bad' based on their performances in terms of the three criteria discussed above. In total, 8 and 6 simulations were evaluated from CORDEX-SEA and CORDEX-EA, respectively. For December event definition, 5 from SEA and 4 from EAS passed the evaluation. For June-December, 4 from each ensemble passed (see tables B1 and B2, figures B2-B5). Only the models that passed the evaluation tests are included in the following analysis.

## 3.5 Meteorological hazard synthesis and conclusions

The influence of anthropogenic climate change on the rainfall from Odette is estimated, for each event definition, by calculating the probability ratio (for an event of equivalent intensity) and the change in intensity (for an event of equivalent likelihood) using a combination of observations and reanalysis datasets, and climate models. The aim is to synthesise results from models



that pass the evaluation along with the observations-based products, to give an overarching attribution statement considering
different kinds of data sources. Figs. 8 & 9 show the changes in probability and intensity for the observations (blue) and models
(red). The dark blue bar shows the synthesised results of the observation-based products. The dark red bar shows the model
synthesis, consisting of a weighted mean using the (uncorrelated) uncertainties due to natural variability. Observation-based
products and models are combined into a single result by computing the weighted average of models (dark red bar) and

observations (dark blue bar): this is indicated by the magenta bar. The white bar around the magenta bar indicates the
unweighted average of uncertainty bounds of the model and observational estimates.

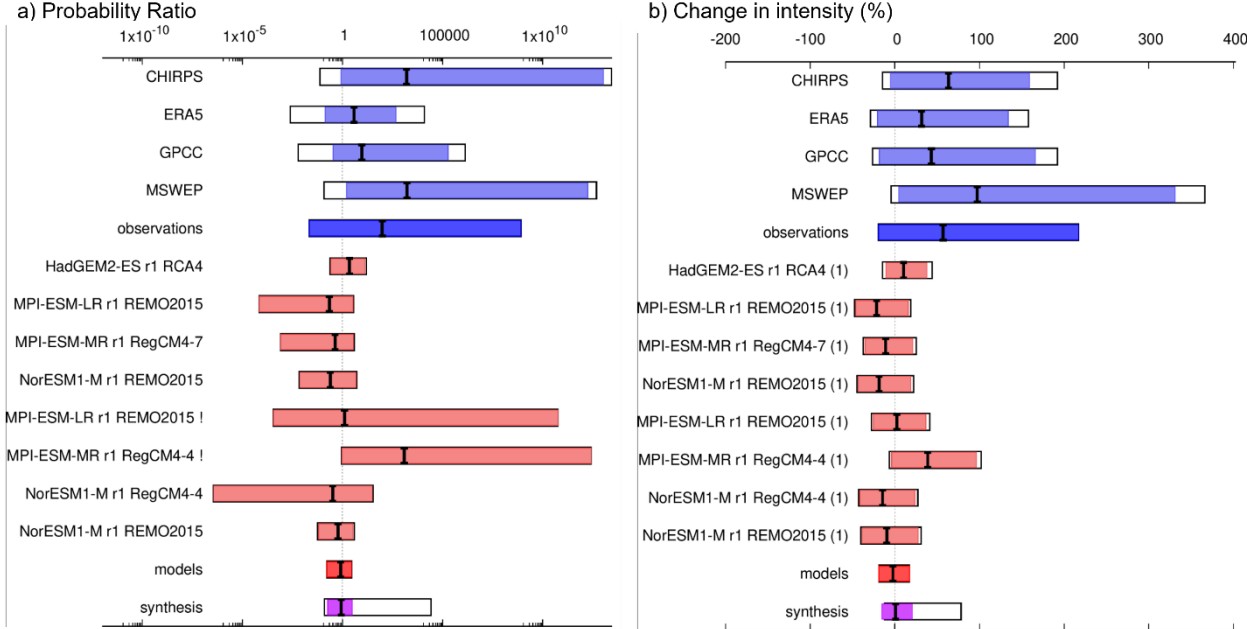

*Figure 8: Synthesis of (a) probability ratios and (b) intensity changes when comparing the 1-day maximum December*
*precipitation that occurred in 2021 with a 1.2 °C cooler climate.*





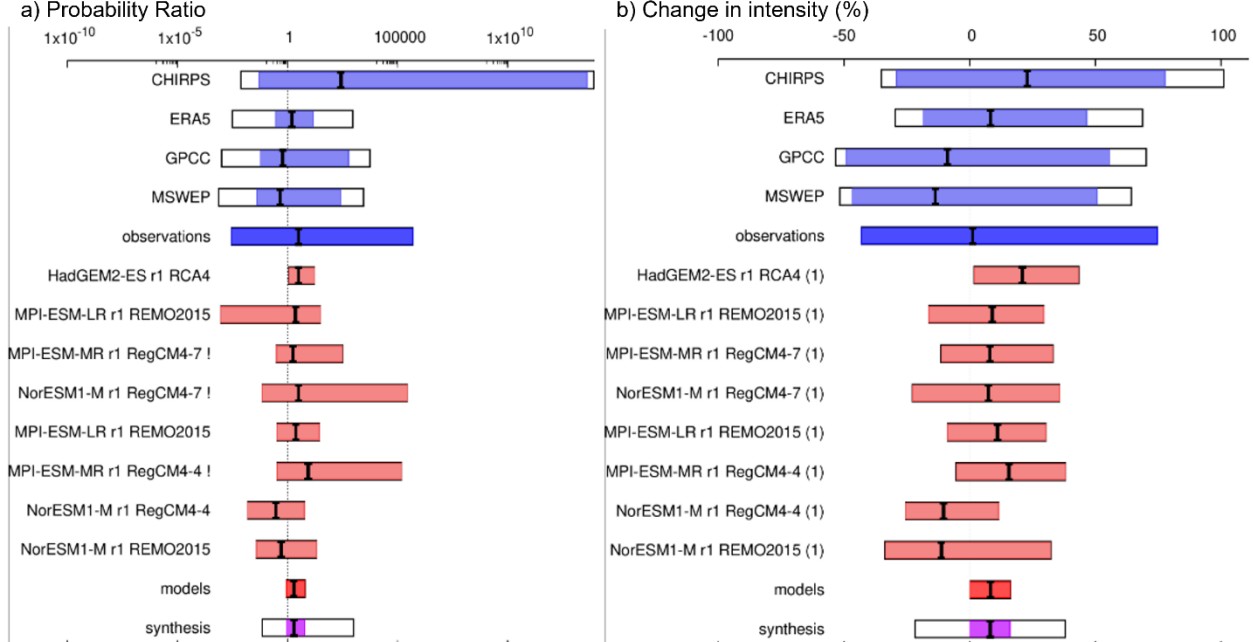

*Figure 9: Synthesis of (a) probability ratios and (b) intensity changes when comparing the 1-day maximum June-December*
*precipitation that occurred in 2021 with a 1.2 °C cooler climate.*

| Time period (data type) | | Probability ratio | Intensity change (%) |
|---|---|---|---|
| **Past-Present** | Observations | 98.6 (0.022 - $10^9$) | 57.0 (-19.2 – 217) |
| | Models | 0.82 (0.17 – 2.92) | -2.30 (-18.8 – 17.4) |
| | **Synthesis (weighted)** | 0.87 (0.18 – 3.12) | 0.87 (-15.5 – 20.5) |
| | **Synthesis (unweighted)** | 8.98 (0.13 – $10^4$) | 27.35 (-12.2 – 78.2) |
| **Present-Future** | **Models only** | 1.03 (0.72 – 1.32) | 0.39 (-3.51 – 3.68) |

*Table 4: Summary of results for events like Typhoon Odette in the December period, both past-present (due to 1.2 °C of*
*warming), and with further warming of 0.8 °C up to a total of 2 °C. Increases (decreases) in likelihood and intensity are*
*highlighted in blue (orange); light for non-statistically significant (95% confidence levels overlap with 1 for PR and 0 for*
*intensity change), and dark for significant.*




| Time period (data type) | | Probability ratio | Intensity change (%) |
|---|---|---|---|
| **Past-Present** | Observations | 3.28 (0.003 - ~$10^6$) | 1.17 (-43.1 - 74.7) |
| | Models | 1.96 (0.90 - 6.05) | **8.29 (0.27 - 16.2)** |
| | **Synthesis (weighted)** | 1.97 (0.91 - 6.05) * | **8.16 (0.22 - 16.0) *** |
| | **Synthesis (unweighted)** | 2.54 (0.07 - 1010) | 4.73 (-21.7 – 37.9) |
| **Present-Future** | **Models only** | 1.55 (0.76 - 3.16) | 4.43 (-2.84 - 12.1) |

*Table 5: Summary of results for events like Typhoon Odette in the June-December period, both past-present (due to 1.2 °C of warming), and with further warming of 0.8 °C up to a total of 2 °C. Statistically significant changes (95% confidence levels overlap with 1 for PR and 0 for intensity change) are in **bold**. *The main results reported for this section.*

For extreme rainfall events in December only, the synthesised observations and climate models respectively give differing trends related to anthropogenic warming. While observations estimate an increase of around a factor of 100 (0.02 – $10^9$) in the likelihood and of 57% (19 – 217%) in the intensity of such events, climate models suggest a decrease of approximately 18% (-83 - +192%) in likelihood and 2% (-19 - +17%) in intensity (table 4). The synthesised result drawing on both lines of evidence gives an approximate 13% (-82 - +212%) decrease in the likelihood and of such events, though a slight increase in intensity

of 1% (-15 - +21%). This synthesis is weighted more heavily upon the model estimate due to the smaller statistical uncertainty on this estimate. This weighting is most informative when quantifying the attributable signal. However, due to the strong disagreement on the qualitative change between the two lines of evidence, it is customary to also consider the unweighted synthesis result, giving equal weight to observations and models, in determining the likely direction of change. The unweighted result shows a strong increase in likelihood, by a factor of 9 (0.13 – 10000), and in intensity, by 27% (-12 - +78%), though the

numbers themselves are largely illustrative given the wide uncertainty ranges. Overall, given the strong increase in observations relative to models, it is more likely than not that anthropogenic warming has resulted in an increase in such extreme December rainfall events, though there is very significant uncertainty on quantifying this effect.

Confidence is higher for extreme rainfall events occurring across the wider TC season of June-December (table 5). Specifically,

at present levels of anthropogenic-induced warming compared to preindustrial times, the best estimate of the change in likelihood for events of similar intensity to Odette in June-December is a factor of 2 (0.9 - 6), and an increase in intensity of events of similar likelihood (1 in 20 years) to Odette in June-December are around 8% (0.2% - 16%) (table 5). Given the





agreement between observations and models, confidence is high in reporting these weighted estimates rather than the more conservative unweighted results and associated range, though this also suggests a strong increase.


We can combine this evidence with that from other attribution studies, as well as the estimates of further intensification with future warming (Tables 4 & 5, Figures B6 & B7), and physical understanding of intensifying short duration precipitation extremes in a warmer atmosphere. Collectively, this evidence suggests that extreme rainfall associated with typhoons such as Odette across the typhoon season striking the southern central Philippines are becoming more frequent due to ACC (high

confidence), and the likelihood of extreme rainfall such as that equal to or exceeding that from Odette has doubled due to ACC.

Clearly, accurately quantifying the changes in extremes is challenging even when the direction of change is known, given the high uncertainties. The results presented in this section are mostly not statistically significant at the 95% level for a range of reasons. Primarily, the shortness of data records. The gridded products used here mostly begin around 1980, except for ERA5,

which naturally leads to higher sampling uncertainty and requires a greater extrapolation of the trend with GMST. In addition, the relative scarcity of in-situ weather stations means that satellite data is relied upon more in the region. Furthermore, the limited accessibility of individual weather station data series has directly hindered the ability of this study to validate the products used. The high uncertainty on the observational estimates is especially prevalent for June-December in which individual observations disagree on the sign of change. In this situation, the longest running dataset, ERA5, gives an increase

across both metrics, increasing our confidence in the synthesised result of an overall increase.

Additional work using complementary methods is required to assess changes in intensity and likelihood of such storms alongside the work presented in this section. For instance, accounting for sources of natural variability by incorporating additional covariates into the statistical models used, exploring changes during the wider 'less active season' from November to March in which Odette occurred, and incorporating other factors such as the anthropogenic influence on the sea surface

temperatures over which the storm developed. Nonetheless, consistent increases in these metrics using a multi-model multi-method statistical attribution framework and across two event definitions suggest that the extreme precipitation associated with Odette has become about twice as likely (with high confidence) due to anthropogenic climate change. This aligns with very well-understood thermodynamic arguments about increasing intensity of extreme rainfall with warming, and with other results attributing TC rainfall in both the WNP and NA basin across more than 10 separate studies using a range of methods.

**4 Attribution of extreme wind speeds using a synthetic tropical cyclone model (IRIS)**

Assessing changes in tropical cyclone (TC) intensity due to climate change remains a challenge for several reasons. First, observational records are only reliable since approximately 1980, and underwent a step-change in the satellite era. Second, TCs are rare in any given location. Third, modelling the wind speeds of the most intense cyclones requires high spatial





resolution (< 10 km), which are computationally expensive and thus ensembles of such models are too small to capture changes
in rare events. To overcome these challenges, here we use a new global tropical cyclone wind model (IRIS, Sparks & Toumi,
2024; Sparks & Toumi, 2025).

This model is based on several robust principles in TC theory. The lifetime maximum intensity (LMI) and where it occurs is
a key determinant of the wind speed at landfall. The LMI is governed by atmospheric conditions through the potential intensity
(PI), which gives a theoretical maximum intensity for a TC based on the atmospheric temperature and humidity vertical
profiles, and sea surface temperatures. Observed historical TC tracks between 1980 and 2021 from the IBTrACS database are
stochastically perturbed by the model. Explicit model description and additional details can be found in the IRIS model
description paper (Sparks and Toumi, 2024).

Two of the main challenges in assessing tropical cyclone risk are the infrequency of landfalling tropical cyclones (TC) and the
short period of reliable observations. Synthetic tropical cyclone datasets can help overcome these problems. As a third method,
we explore this using a new global tropical cyclone wind model (IRIS, Sparks & Toumi, 2024; Sparks & Toumi, 2025) with
several key innovations. Foremost, it recognises that the key step for estimating landfall wind speed is the location and value
of the lifetime maximum intensity (LMI). It redefines the problem as one of decay only. The initial intensity, life-time
maximum, is assumed physically constrained by the thermodynamic state as defined by the potential intensity (PI).

Potential intensity is a well-established concept in tropical cyclone theory that seeks to define an upper limit of the maximum
wind speed. This upper limit can be diagnosed from the sea surface temperature and the vertical profiles of the humidity and
temperature. Observations show that the relative intensity, defined as observed maximum intensity divided by the potential
intensity, follows a robust uniform distribution. This drives the stochastic model lifetime maximum intensity. The landfall
intensity is then a fraction of this lifetime maximum intensity depending on the time to landfall. Tracks are based on IBTrACS
observations. Further details can be found in the IRIS model description paper (Sparks and Toumi, 2024). IRIS calculates
basin and landfall wind speed intensity distributions from the location of LMI and the corresponding PI at that location, based
on observed tracks between 1980 and 2021.


At the time of Odette, the global mean temperature was ~ 1.2 °C above pre-industrial temperature (Fig. 10). The anthropogenic
trend is assumed to be the global zonal mean PI trend, removing potential model biases and regional-scale variability, and use
the observed PI trend since 1979 from ERA-5. To estimate the pre-industrial potential intensity state, we extrapolate backwards
the current observed trends from 1979 to present. This approach avoids the need for any specific climate model and is therefore
both simple and robust. Figure 10 shows the global mean surface temperature time series we use to scale to the pre-industrial
PI. Figure 11 shows the global zonal mean PI change for the region near the Philippines.



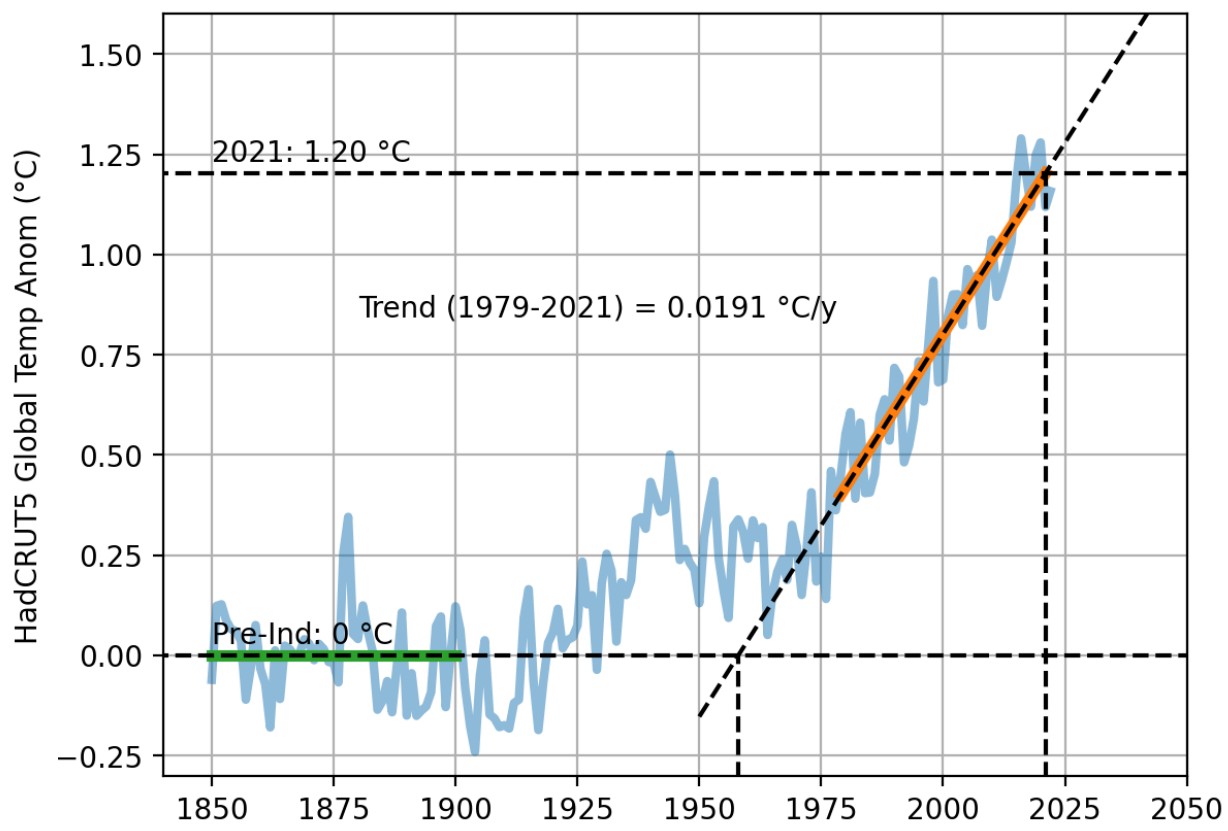

*Figure 10: Global mean surface temperature showing scaling method. The 2021 warming is +1.20 above pre-industrial, which*
*is defined as the period 1850-1900. The warming trend between 1979-2021 (the ERA5 period) is +0.0191 °C/yr.*



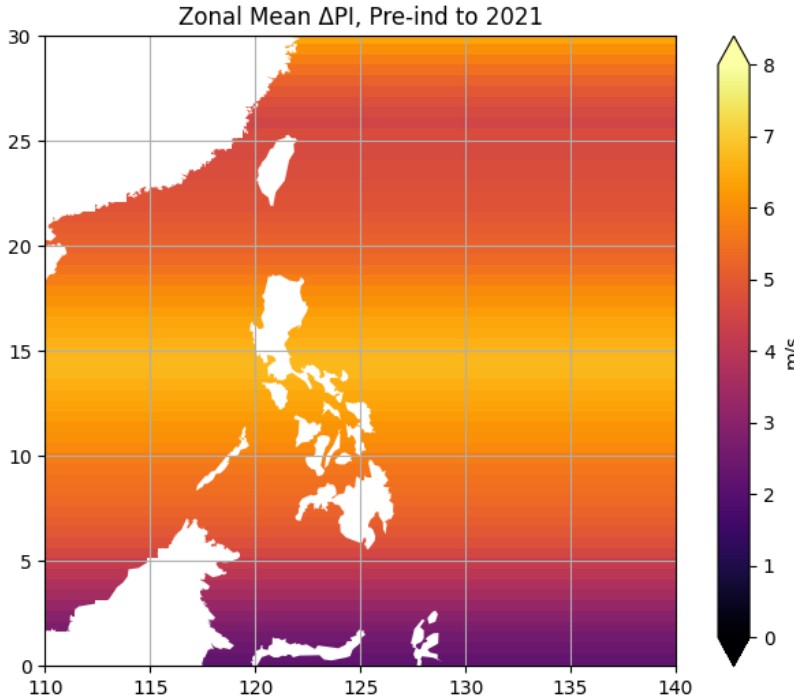

*Figure 11: Zonally averaged potential intensity change from the pre-industrial baseline to 2021 over the western Pacific region using ERA5.*



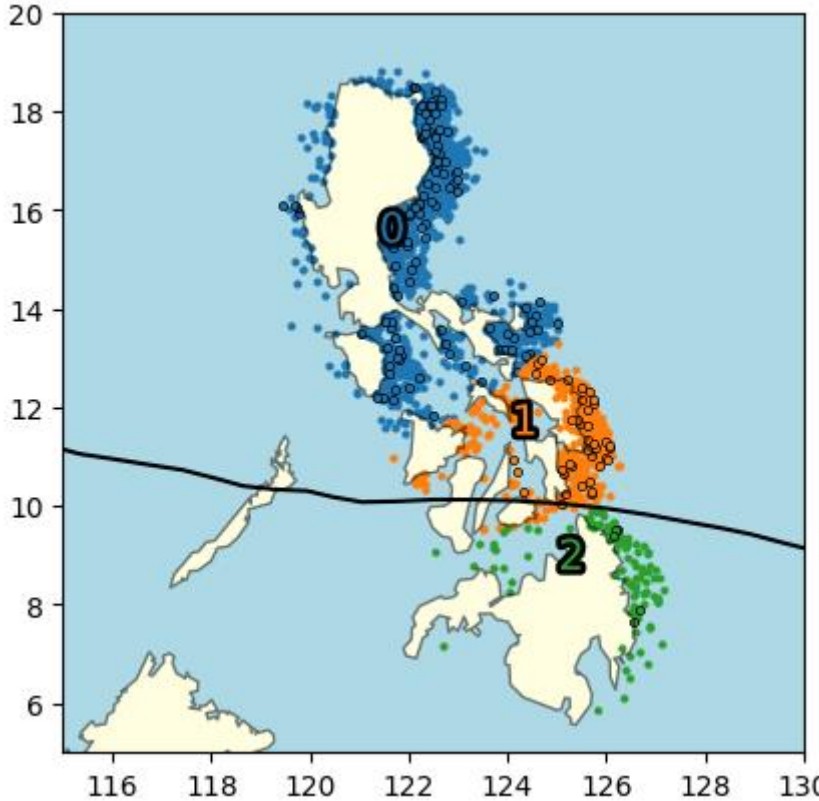


*Figure 12: Observed (black circles) and IRIS (coloured dots) landfall events in the Philippines. Observations are IBTrACS (1980-2021). IRIS are from a sample of 420 years. Landfall events are categorised by location into north (blue, 0), central (orange, 1) and south (green, 2). Black line shows the path of Odette.*

We split the Philippines into three "Gates" based on the landfall climatology: North (0), Central (1) and South (2). Odette made landfall at the southern edge of zone 1 (Figure 12). The numbers of landfalling events and their intensities at landfall are tallied, enabling the construction of return curves (Fig. 13). From this, the likelihood of a landfalling event at category 5 can be estimated under both current and preindustrial conditions. This in turn enables estimation of the fractional attributable risk (FAR), given by


$$FAR = 1 - \frac{P_0}{P_1},$$

(1)

Where $P_1$ and $P_0$ are the probabilities of event occurrence in the current and preindustrial climates, respectively.



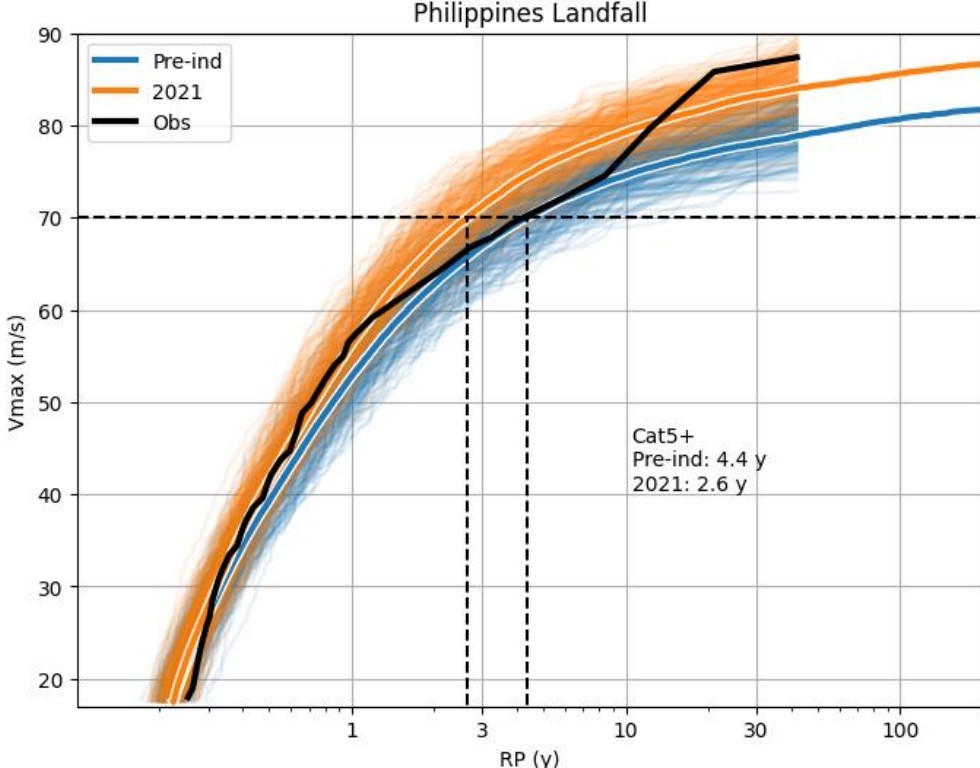

*Figure 13: Return curves for all-Philippines landfall events. IBTrACS 1980-2021 observations (black) and 10,000-year IRIS simulations in pre-industrial (blue) and 2021 (orange) climate conditions. Ensembles of 42-year samples of IRIS output shown in fine lines. Dotted lines show return periods of Cat 5 events in IRIS simulations.*

For the all-Philippine landfall for a Category 5+ wind speed, IRIS estimates a return period of 2.6 years in 2021 compared to a pre-industrial value of 4.4 years (Figure 13). This corresponds to an increase in likelihood of approximately 70%, or equivalently a FAR of about 0.39 (equation 1). In addition to this probabilistic framing, there is an equivalent interpretation of this analysis on the attributable change in intensity. The wind speed of a typhoon with a return period of 2.6 years has increased by approximately 5 m/s from pre-industrial to 2021. This result agrees with other attribution analysis for typhoons in the region using different approaches (Delfino et al., 2023). This wind speed increase is close to the uncertainty of measurements and would not be detectable through observational analysis alone.

If we instead consider the return periods at the three Gates 0,1 and 2, then the FAR is 0.43, 0.34, and 0.28, respectively. The increased risk with increased latitude can be understood by the similar latitude pattern of PI change (Fig. 11). Given the larger sample size for the all-Philippine landfall dataset, this is the most robust FAR result i.e. about 0.4. For the more specific location at Odette's landfall, IRIS suggests that a fractional attributable risk of about 0.3 (between 0.34 and 0.28, Gate 1 and





2 respectively) may be reasonably considered, but is not as statistically robust as the all-Philippine result. Given that it is always possible to specify with greater granularity the specific event domain (e.g. tailoring the 'Gate' to an area just around the exact landfall), selecting an event definition is a compromise between relevance to the event itself and statistical power gained from a larger sample size. We conclude that the all-Philippines region (with FAR ~ 0.4) is the best compromise between these factors, as it provides both a large sample of events and risk-relevant information for storms like Odette across the nation. The current return period in the individual gates is in the range of 14 to 41 years, which is comparable to that found for rainfall (Table 5). To give an uncertainty range on the estimates of FAR, independent 120-year samples are drawn from the wider 10,000-year simulation. This enables determination of the probability distribution of FAR (Fig. 15), returning a 95% confidence interval of 0.03 to 0.66. The standard error of the best estimate mean, 0.39, is 0.015, or less than 4%.

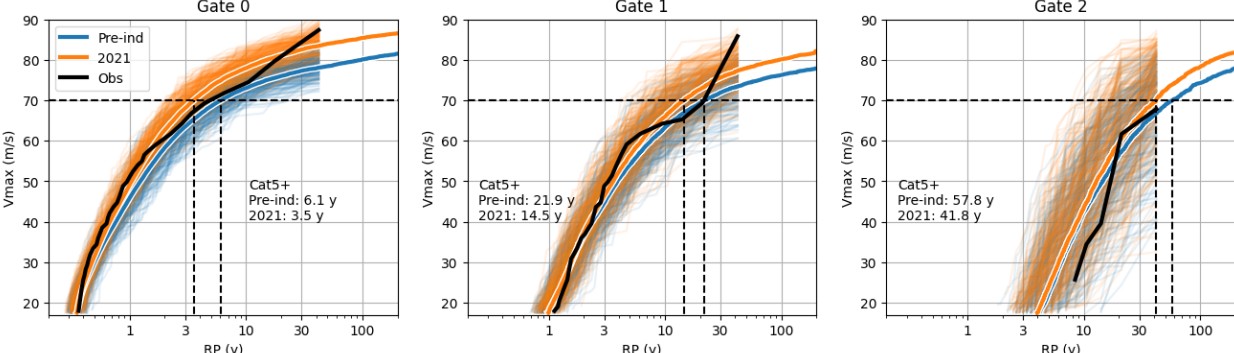

*Figure 14: As in Figure 13 but for regions of Philippines shown in Figure 12.*




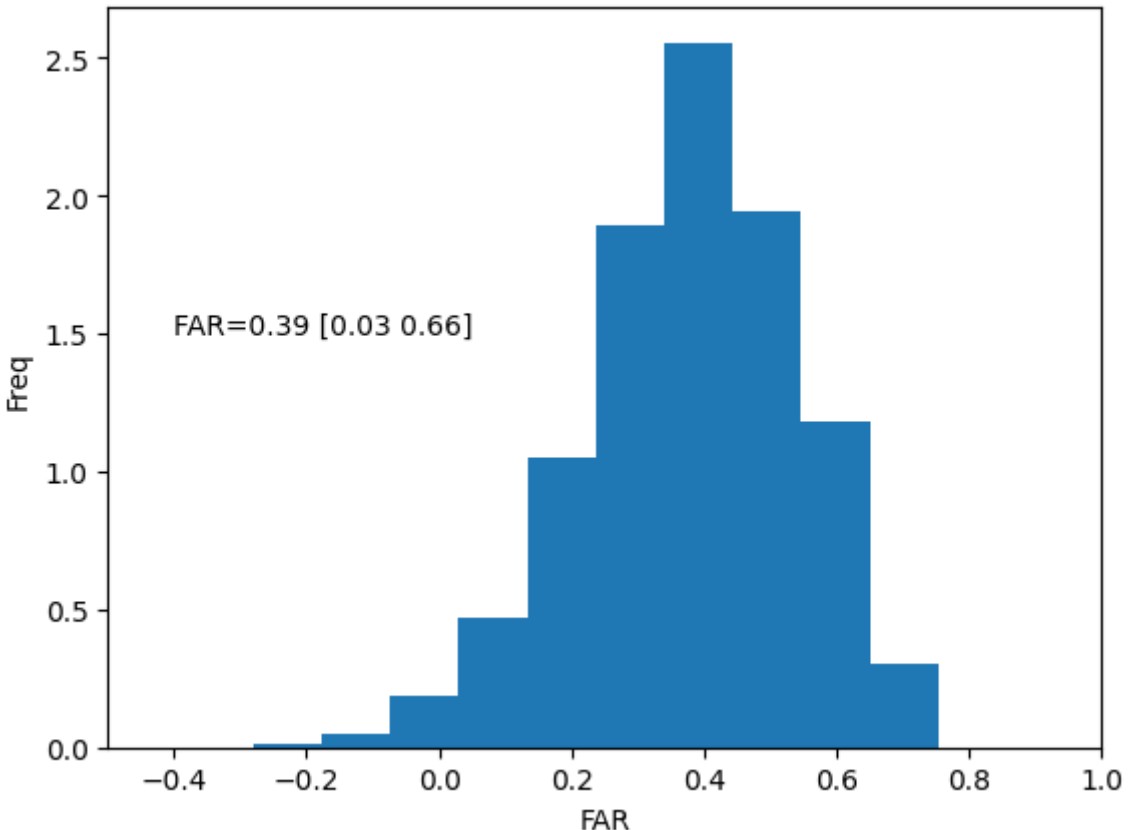

*Figure. 15: Probability distribution of 120-year samples of the fractional attributable risk in the 10,000-year simulation,*
*including the 95% confidence intervals on the best estimate FAR value.*

**5. Discussion and conclusions**

In this section, the results of meteorological hazard attribution are summarised and the implications of this for the damages from Typhoon Odette that are attributable to ACC are discussed. In particular, the FAR statistic (Sect. 4) is employed based on the hazard assessments in sections 3 & 4. Interpretation of this estimate involves several caveats and therefore requires care.

The multiple meteorological hazards from Odette studied here each produced one or more probability ratios due to anthropogenic climate change. For instance, in Sect. 3, extreme rainfall totals from TCs like Odette have become about 2 times (or 100%) more likely, with an uncertainty range of 0.9-6 (10% less to 500% more), due to anthropogenic climate change. In Sect. 4, the wind speeds from category 5 TCs were found to occur around 70% more often, with an uncertainty range of 3 - 200% more, due to anthropogenic climate change. Using equation 1, this gives best estimate values of FAR of 0.49 (-0.099 –
0.83) for rainfall and a FAR of 0.39 (0.03 – 0.66) for wind speed. Table 6 shows this range of values across all hazards, event definitions and synthesis methods, calculated using the best estimates and lower and upper bounds of the probability ratios.





| Hazard (event definition) | | FAR (95% confidence interval) | | |
|---|---|---|---|---|
| | | Best estimate | Lower bound | Upper bound |
| Rainfall | June-December (weighted) | 0.49* | -0.099 | 0.83 |
| | June-December (unweighted) | 0.61 | -13.29 | ~1 |
| | December (weighted) | -0.15 | -4.56 | 0.68 |
| | December (unweighted) | 0.89 | -6.69 | ~1 |
| Winds | All-Philippine | 0.39* | 0.03 | 0.66 |
| | Gate 0 | 0.43 | | |
| | Gate 1 | 0.34 | | |
| | Gate 2 | 0.28 | | |

*Table 6: Estimates of the FAR for each hazard associated with Odette, for each event definition and synthesis method, alongside uncertainties. *The best estimates for each hazard.*

## 5.1 Using the Fractional Attributable Risk

Using the FAR to attribute damages from an event requires careful interpretation and involves several conditions. First, the FAR statistic is asymmetric in nature, ranging between an upper bound of 1 (an event was impossible without ACC), to negative infinity (an event made impossible by ACC). The central estimate of 0 represents no change due to ACC, positive values (0-1) showing a quantifiable increase due to ACC and negative values ($< 0$) show a quantifiable decrease.

For rainfall, all FAR results are highly uncertain as the 95% confidence interval encapsulates no change in likelihood (FAR=0), making them statistically insignificant (Table 6). However, statistical significance alone is not sufficient to dismiss a hypothesis in the presence of other relevant evidence, and an overreliance on these uncertainty ranges risks type II errors in which the influence of climate change is falsely dismissed (Shepherd, 2019). Given the relatively short length of data records, it is expected that natural variability would result in a wide uncertainty range, especially for the December-only event definition (which includes plausible best estimate probability ratios of both 9 and 0.9). Furthermore, as shown in Sect. 3, multiple event





definitions give similar results and this result aligns with physical understanding, both of which reinforce confidence in the accuracy of these results. Consequently, confidence is high that anthropogenic climate change intensified the rainfall of Odette, with the best estimate yielding a doubling of the likelihood of such intense rains. Meanwhile, confidence is high both quantitatively and qualitatively that ACC increased the wind speeds of Odette, as the all-Philippines result is statistically
significant, best estimate results are positive across all 'Gates', and the underlying method is based on physically robust potential intensity theory.

Second, the FAR is probabilistic in nature. This statistic has been used in several attribution studies, including TCs and other extremes, to estimate attributable damage. For example, rainfall from Typhoon Hagibis was increased in likelihood by 15-150%; the resultant best estimate FAR of 0.4 led to an estimate of US$4 bn in attributable damages from the total damages of
US$10 bn (Li and Otto, 2022). Hurricane Harvey rainfall was found to have increased in likelihood by a factor of around 4 across multiple studies, giving FAR=0.75 and an estimate of attributable damage of US$67 bn (Frame et al., 2020).

The key message in such cases is that the "[probability] ratio can be interpreted as a lower bound to relative changes in the expected losses due to the extremes provided that the consequences of extreme events of a fixed intensity do not decrease with warming, consistent with expectations" (Kharin et al., 2018). In essence, over a long period at the current level of warming,
the losses due to events of at least the magnitude of the event in question would be a factor of the probability ratio larger than those due to the same class of events in preindustrial times. Using equation 1, the FAR is then used to quantify attributable damage related to this change in risk of the most extreme events.

More broadly, the FAR is valid for a class of events to which the event belongs, which also includes events at least as intense as that event. In general, it is not a direct attribution of damage from a single event but attributes a level of impact from events
of at least the given intensity. The other way to attribute damage from specific extremes within the same framework involves using a relative magnitude framing (Perkins-Kirkpatrick et al., 2022). Within this method, the change in magnitude due to ACC for an event of equivalent frequency is calculated. This is then fed into a 'damage function', returning a level of damage done by events of the same probability in different climates, giving the 'direct' attributable impacts.

The latter framing was not used here for several reasons. Primarily, constructing a damage function (or 'vulnerability curve')
from the available disaster data introduces an additional layer of uncertainties; in a recent study that empirically constructed such functions, the uncertainties in this region were particularly large (Eberenz et al., 2021). There is also significant variation in the damage susceptibility across the Philippines, rendering national-scale assessments difficult (Baldwin et al., 2023). It is noteworthy that the eastern Visayas have a significantly larger damage fraction than both the national capital region (surrounding Manila) and the national average for a given wind speed (Baldwin et al., 2023).




While debate continues as to the usefulness of FAR in quantifying the attributable impacts of a specific extreme event, there is a special case for the interpretation of FAR-based impact statements when such events are 'system-breaking'. These events are above a key threshold in magnitude at which impacts rapidly increase. For instance, this could involve the rainfall required for a river to burst its banks, or the wind speeds exceeding the strength of local structures. In this case, the FAR-based approach gives a more direct estimate of damage due to ACC because the frequency of such events is all that determines the total level

of impacts. While it is not clear if Odette was 'system-breaking', and the highly uncertain vulnerability curves described above do not inform this, category 5 TCs are known to produce winds beyond the capacity of many structures to withstand. In many regions in which vulnerability curves are more confident, the damage fraction is significantly above 50% for winds of category 5 level (Eberenz et al., 2021). Put simply, for the most intense storms an increasing frequency likely causes an equivalent increase in damage, while comparably small changes in overall magnitude may be of relatively little importance.

As a result of the above arguments, this study uses the FAR-based framing, as it is grounded in the simple logical assertion that more intense and probable hazards result in greater damage than less intense and probable hazards, for a given vulnerability and exposure context. While not tied directly to the magnitude shift of this specific storm, it is part of the same statistical framing and carries an important attribution statement. That is, events at least as intense as Odette, and therefore most likely causing at least as much damage, have become more likely by a given factor (the probability ratio) due to anthropogenic

climate change. And, consequently, an equivalent fraction of the damages of all events within that 'class' can be attributed to its influence through the FAR.

## 5.2 Attributable Impacts of Typhoon Odette

TCs like Odette are compound events. The impacts of TC systems often result from a combination of several meteorological extremes. This includes rainfall that may lead to flash and riverine flooding and mass movements such as landslides and

mudslides, and extreme winds that lead to direct destruction of property and infrastructure as well as indirect damage through causing storm surges, which further lead to coastal flooding. During a TC, it is the combination of these factors that lead to such devastation, as each heightens the regional vulnerability to other concurrent hazards, resulting in a non-linear amplification of impacts. The assessments conducted in sections 3 and 4 are for rain and wind hazards in isolation. However, since both are increasing in parallel, it is likely that their combined attributable increases made the damages disproportionately

larger than either their individual contributions or their sum.

This has two implications for this study. First, it reinforces the choice to use a FAR based approach to damage attribution given the insurmountable challenge not only in constructing vulnerability curves, as described above, but in disentangling the relative role of each type of hazard. Second, in line with compound events attribution theory, the influence of ACC on the combined impacts of Typhoon Odette – the bivariate FAR – is a function of both its influence on each of the hazards in isolation, given

by the univariate FARs, and the correlation between hazards (Zscheischler and Lehner, 2022).





Within this study, it is impossible to directly assess the joint changes in rainfall and high winds because they are modelled using different methods. For many compound events, such as concurrent heatwaves or droughts in different breadbasket regions, or individual droughts consisting of both heat and a lack of rainfall, it is not always intuitively clear how strongly the phenomena are correlated. However, it is clear that for TCs the two phenomena in question are highly correlated. This was

tested using the ERA5 and MSWEP/X datasets (Fig. C1). Within these datasets, the June-December maximum rainfall used in the trend analysis in section 3 and the maximum and average wind speeds on the same dates have a Pearson correlation coefficient of ~0.64 (p value approximately $10^{-10}$), confirming that there is indeed a strong positive correlation. That in turn suggests that the following criterium, derived empirically through a case study using several climate models, is met in this case: "if variables are strongly correlated and have similar trends, the bivariate FAR is at the upper range of the univariate

FARs" (Zscheischler and Lehner, 2022).

This means that the overall FAR for the combined extremes of Odette, which is most pertinent to the damages experienced, is most likely equal to or slightly greater than that for the larger of the univariate FAR statements. This gives a best estimate of FAR >= 0.49 related to the compounding winds and rainfall Odette, suggesting that the risk of similar impacts has likely more than doubled due to ACC. However, without further work within a consistent modelling framework it is challenging to be

precise or to quantify the uncertainty on this estimate. For risk assessment and adaptation, the individual probability ratios (or FARs) are more robust, though their combined effects and the potentially compounding effect of ACC should not be neglected.

Bringing together the many lines of evidence presented in this study, we conclude that ACC has likely more than doubled the risk of combined high winds and heavy rainfall like those due to Typhoon Odette. Using the simple economic damages reported by EMDAT, this suggests damages due to ACC from Odette exceed US$450 million (in 2021). This conclusion is based on

careful consideration of a range of quantitative and qualitative analyses. Finally, it is likely a conservative estimate of overall damages when considered against the many forms of damage, suffering and disruption experienced by the people of the Philippines as a result of Odette.

**Competing Interests**

The authors declare no competing interests.



# Appendices


## A. Section 2 figures

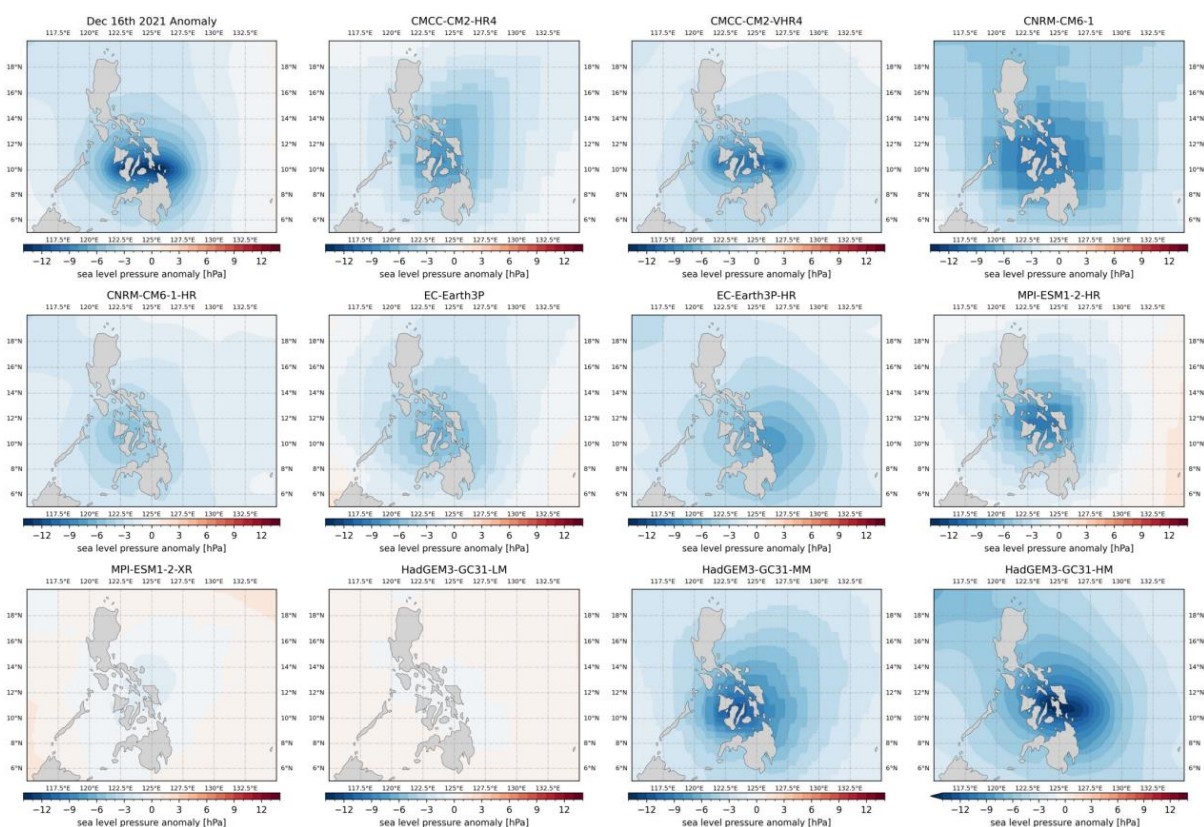

*Figure A1: For each model, the sea level pressure anomaly patterns over (5-20 °N,115-135 °E), with the highest spatial correlation to Dec 16th 2021 during Odette landfall is shown.*


## B. Section 3 figures

### B.1.1 Observational attribution






*Figure B1: Statistical fits and trends in the four observational datasets, ERA5, CHIRPS, MSWEP and GPCC for December.*
*a) Shows the GEV fit to the ERA5 data at two levels of the covariate GMST: in 2021 (red line) and in a 1.2 °C cooler climate*
*(blue line). The purple line shows the magnitude of Odette. b) Shows the estimated trend, with the event associated with Odette*
*highlighted in purple. c) and d), e) and f), g) and h): As in (a) and (b) for CHIRPS, MSWEP and GPCC data.*





**B.1.2 Model evaluation**

| Model / Observations | Seasonal cycle | Spatial pattern | Dispersion | Shape parameter | Event magnitude (mm) |
|---|---|---|---|---|---|
| CHIRPS | | | 0.336 (0.196 … 0.43) | -0.147 (-1.051 … 0.548) | 83.26 |
| ERA5 | | | 0.566 (0.479 … 0.626) | 0.065 (-0.193 … 0.348) | 76.48 |
| GPCC | | | 0.363 (0.26 … 0.433) | 0.027 (-0.255 … 0.29) | 78.94 |
| MSWEP | | | 0.465 (0.359 … 0.55) | -0.07 (-0.599 … 0.124) | 76.98 |
| **Models** | | | | | **Threshold for 20-year return period event (mm)** |
| **CORDEX-SEA** | | | | | |
| CNRM-CM5_r1_RCA4 () | bad | good | 0.453 (0.356 … 0.539) | -0.197 (-0.70 … 0.072) | 76.32852 |
| HadGEM2-ES_r1_RCA4 () | good | reasonable | 0.473 (0.345 … 0.565) | -0.077 (-0.291 … 0.135) | 87.81494 |
| HadGEM2-ES_r1_RegCM4-7 () | reasonable | bad | 0.275 (0.179 … 0.32) | 0.126 (-0.107 … 0.839) | 38.63236 |
| HadGEM2-ES_r1_REMO2015 () | reasonable | bad | 0.402 (0.302 … 0.475) | -0.136 (-0.389 … 0.078) | 79.96933 |
| MPI-ESM-LR_r1_REMO2015 () | good | reasonable | 0.379 (0.287 … 0.457) | -0.209 (-0.803 … 0.00043) | 76.04281 |
| MPI-ESM-MR_r1_RegCM4-7 () | reasonable | reasonable | 0.287 (0.219 … 0.340) | -0.138 (-0.523 … 0.308) | 35.30564 |
| NorESM1-M_r1_RegCM4-7 () | reasonable | reasonable | 0.327 (0.258 … 0.459) | -0.114 (-1.13 … 0.138) | 35.02233 |
| NorESM1-M_r1_REMO2015 () | reasonable | reasonable | 0.343 (0.226 … 0.432) | -0.147 (-0.429 … 0.181) | 86.92025 |
| **CORDEX-EAS** | | | | | |
| HadGEM2-ES_r1_RegCM4-4 () | bad | reasonable | 0.484 (0.341 … 0.574) | -0.228 (-0.821 … 0.39) | 23.46256 |
| HadGEM2-ES_r1_REMO2015 () | bad | bad | 0.533 (0.420 … 0.617) | 0.026 (-0.411 … 0.322) | 123.707 |



| | Seasonal cycle | Spatial pattern | Dispersion | Shape parameter | Event magnitude (mm) |
|---|---|---|---|---|---|
| MPI-ESM-LR_r1_REMO2015 () | good | reasonable | 0.461 (0.342 … 0.574) | -0.332 (-0.680 … -0.139) | 103.8969 |
| MPI-ESM-MR_r1_RegCM4-4 () | reasonable | reasonable | 0.404 (0.289 … 0.503) | -0.227 (-0.778 … -0.069) | 46.10828 |
| NorESM1-M_r1_RegCM4-4 () | good | reasonable | 0.310 (0.235 … 0.373) | -0.226 (-0.856 … -0.029) | 25.95332 |
| NorESM1-M_r1_REMO2015 () | reasonable | reasonable | 0.332 (0.244 … 0.392) | 0.062 (-0.185 … 0.274) | 101.2148 |

*Table B1: Model evaluation results for rainfall extremes in December in the study region in the southern Philippines. Each row is a single model, coloured by the evaluation ranking: red for 'bad' and orange for 'reasonable'. No models were ranked 'good' in the evaluation. Models ranked 'bad' are not included in the analysis.*


| Model / Observations | Seasonal cycle | Spatial pattern | Dispersion | Shape parameter | Event magnitude (mm) |
|---|---|---|---|---|---|
| CHIRPS | | | 0.253 (0.183 ... 0.322) | -0.38 (-1.1 ... 0.15) | 83.26 |
| ERA5 | | | 0.289 (0.229 ... 0.331) | 0.079 (-0.11 ... 0.31) | 76.48 |
| MSWEP | | | 0.320 (0.236 ... 0.380) | 0.022 (-0.26 ... 0.32) | 78.94 |
| GPCC | | | 0.285 (0.196 … 0.336) | -0.005 (-0.37 … 0.478) | 76.98 |
| **Models** | | | | | **Threshold for 20-year return period event (mm)** |
| **CORDEX-SEA** | | | | | |
| CNRM-CM5_r1_RCA4 () | bad | good | 0.380 (0.293 ... 0.444) | -0.28 (-0.59 ... -0.077) | 96.01875 |
| HadGEM2-ES_r1_RCA4 () | good | reasonable | 0.307 (0.199 ... 0.375) | -0.062 (-0.48 ... 0.38) | 116.6165 |
| HadGEM2-ES_r1_RegCM4-7 () | reasonable | bad | 0.144 (0.103 ... 0.177) | -0.26 (-0.72 ... 0.14) | 61.15176 |
| HadGEM2-ES_r1_REMO2015 () | reasonable | bad | 0.132 (0.102 ... 0.207) | -0.35 (-1.1 ... 0.025) | 97.84563 |
| MPI-ESM-LR_r1_REMO2015 () | good | reasonable | 0.184 (0.137 ... 0.225) | -0.067 (-0.38 ... 0.17) | 103.6128 |
| MPI-ESM- | reasonable | reasonable | 0.194 (0.145 ... 0.231) | 0.16 (-0.26 ... 0.55) | 75.11458 |





| | | | | | |
|---|---|---|---|---|---|
| MR_r1_RegCM4-7 () | | | | | |
| NorESM1-M_r1_RegCM4-7 () | reasonable | reasonable | 0.250 (0.181 ... 0.325) | -0.32 (-1.1 ... 0.16) | 55.74366 |
| NorESM1-M_r1_REMO2015 () | reasonable | reasonable | 0.153 (0.116 ... 0.183) | -0.20 (-0.51 ... -0.012) | 90.9818 |
| **CORDEX-EAS** | | | | | |
| HadGEM2-ES_r1_RegCM4-4 () | bad | reasonable | 0.182 (0.130 ... 0.265) | -0.26 (-1.1 ... 0.13) | 50.78031 |
| HadGEM2-ES_r1_REMO2015 () | bad | bad | 0.170 (0.121 ... 0.205) | 0.10 (-0.16 ... 0.43) | 137.141 |
| MPI-ESM-LR_r1_REMO2015 () | good | reasonable | 0.184 (0.141 ... 0.216) | 0.042 (-0.29 ... 0.26) | 134.0954 |
| MPI-ESM-MR_r1_RegCM4-4 () | reasonable | reasonable | 0.164 (0.110 ... 0.199) | -0.12 (-0.33 ... 0.12) | 51.68393 |
| NorESM1-M_r1_RegCM4-4 () | good | reasonable | 0.157 (0.120 ... 0.189) | -0.11 (-0.51 ... 0.17) | 36.24989 |
| NorESM1-M_r1_REMO2015 () | reasonable | reasonable | 0.232 (0.171 ... 0.281) | 0.10 (-0.43 ... 0.38) | 129.2698 |

*Table B2: Model evaluation results for rainfall extremes in December in the study region in the southern Philippines. Each row is a single model, coloured by the evaluation ranking: red for 'bad' and orange for 'reasonable'. No models were ranked 'good' in the evaluation. Models ranked 'bad' are not included in the analysis.*





Figure B2: *Seasonal cycles of normalised monthly precipitation for the period 1980-2010 in the ERA5 and MSWEP and CORDEX-SEA models.*







*Figure B3: Spatial patterns of mean daily precipitation over the Philippines for the period 1980-2010 in the three highest-resolution observational and reanalysis products and CORDEX-SEA models. The study region is highlighted in red.*




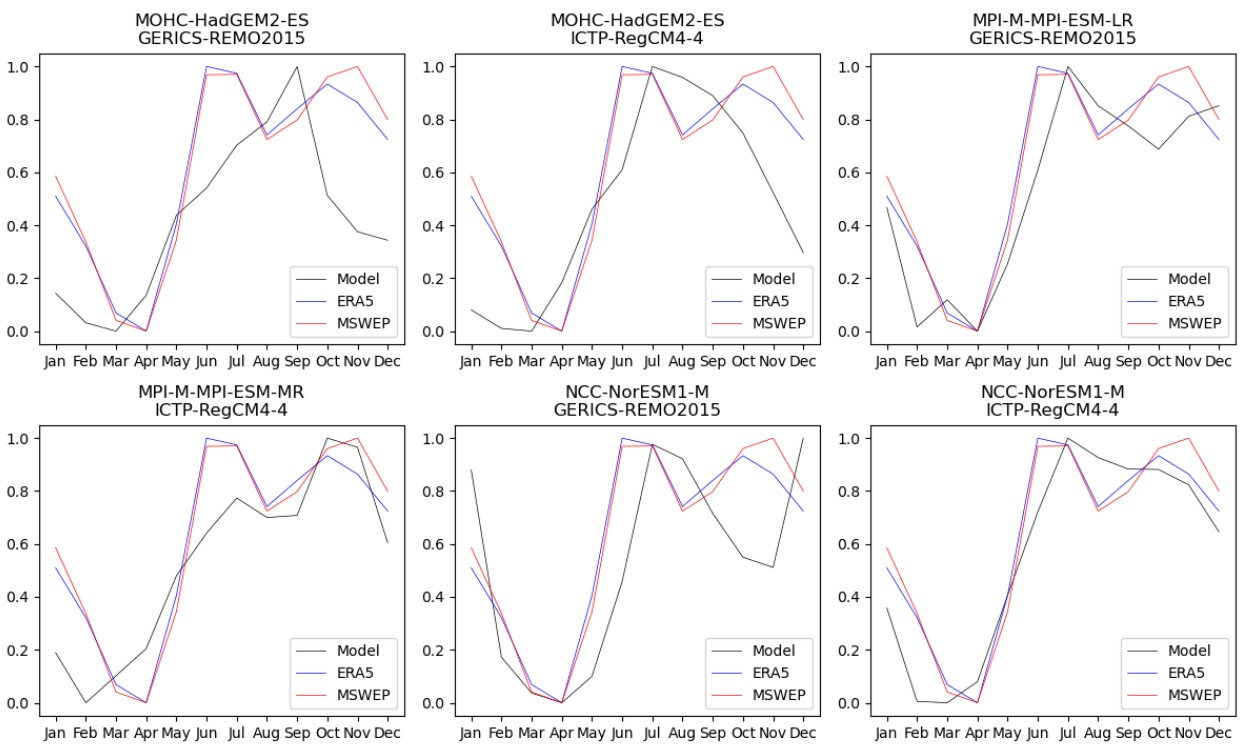

*Figure B4: Seasonal cycles of normalised monthly precipitation for the period 1980-2010 in the ERA5 and MSWEP and CORDEX-EAS models.*






*Figure B5: Spatial patterns of mean daily precipitation over the Philippines for the period 1980-2010 in the three highest-resolution observational and reanalysis products and CORDEX-EAS models. The study region is highlighted in red. Note: The*
*study region is restricted to a lower bound of 8 °N (rather than 7 °N) to minimise boundary effects.*



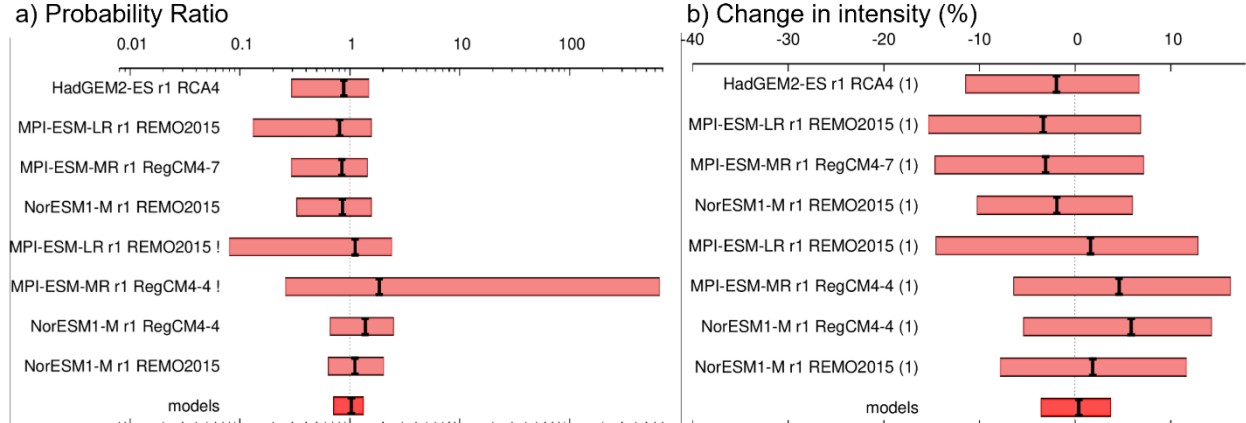

*Figure B6: Synthesis of (a) probability ratios and (b) intensity changes when comparing the 1-day maximum December precipitation that occurred in 2021 with a 0.8 °C warmer climate (for a total of 2 °C above preindustrial levels).*


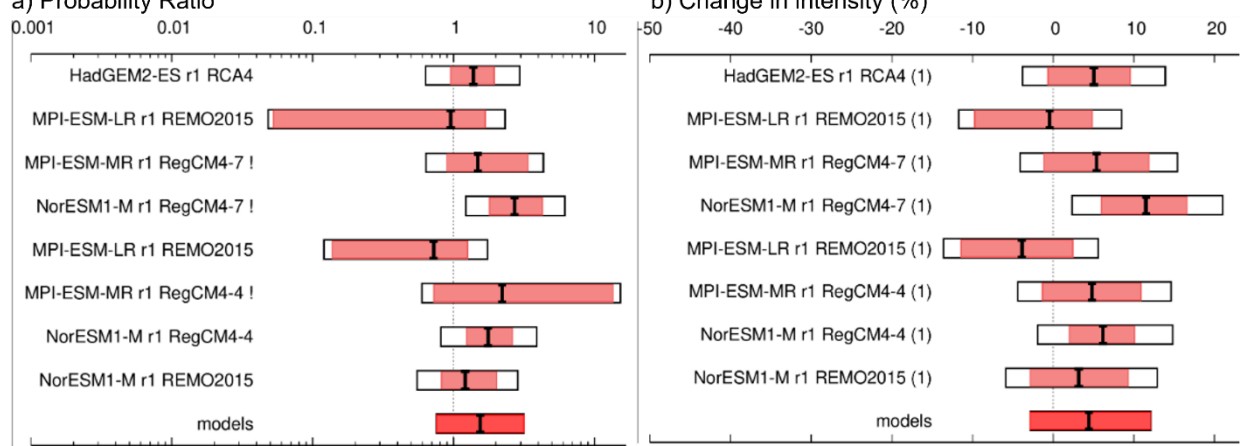

*Figure B7: Synthesis of (a) probability ratios and (b) intensity changes when comparing the 1-day maximum June-December precipitation that occurred in 2021 with a 0.8 °C warmer climate (for a total of 2 °C above preindustrial levels).*




**C. Section 5 figures**

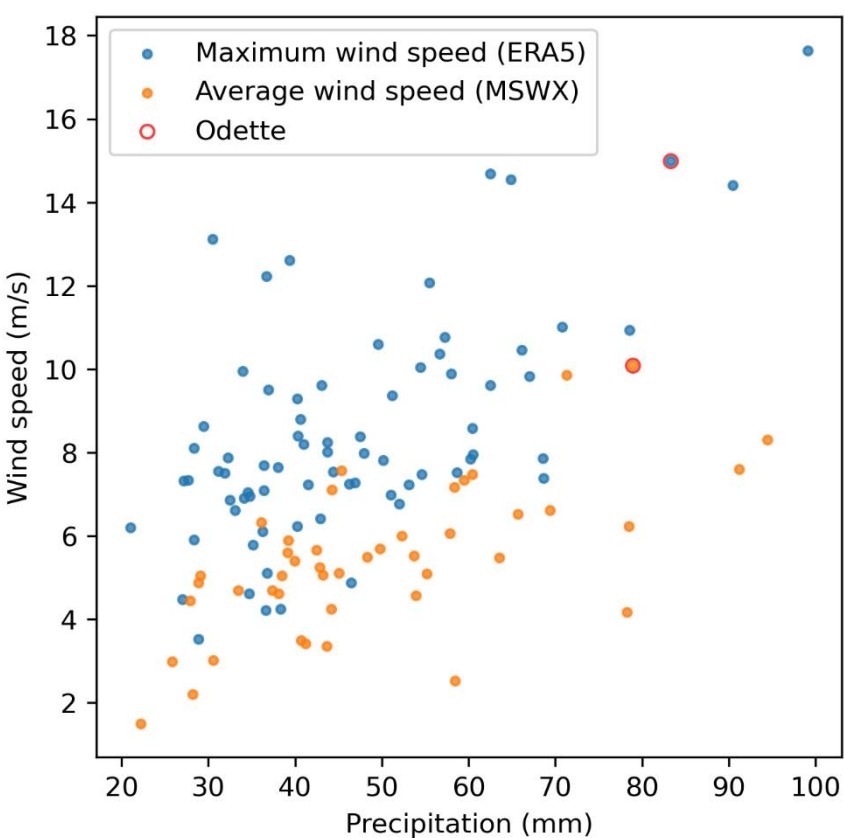

*Figure C1: Joint distributions of extreme daily precipitation used for trend analysis in section 3 with daily maximum wind speeds (blue) and daily average wind speeds (orange). Maximum wind speeds are from ERA5 and average wind speeds are from MSWX, plotted against the June-December maximum precipitation in each dataset used for trend analysis. Typhoon Odette values are highlighted in red.*




## Funding

This research was funded by Uplift. RT and NS additionally received funding from the Singapore Green Finance Center,
Natual Environmental Reseach Council (Grant number: NE/W009587/1) and the Vodafone Foundation.

## Author contribution

All authors were involved in study conception and reviewing the manuscript. SL carried out the analysis and prepared all figures in section 2. BC carried out the analysis and prepared all figures in section 3 and prepared the manuscript with input from all authors. RT and NS carried out the analysis and prepared all figures in section 4.



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
