# Peer review of "The influence of anthropogenic climate change on Super Typhoon Odette (Typhoon Rai) and its impacts in the Philippines"

_EGUsphere, 2025_

## Referee Comment (RC2)

**Review of "The influence of anthropogenic climate change on Super Typhoon Odette (Typhoon Rai) and its impacts in the Philippines" by Ben Clarke et al.**

**General comments**

The study assesses the human-induced climate change effect on Super Typhoon Odette by using three approaches: an analogues-based method, the standard WWA probabilistic approach, and an interesting approach based on an ensemble of TCs generated via the IRIS model. This combination of approaches is the strength of the analysis. Accordingly, in the final discussion, I would appreciate a comprehensive overview that synthesizes all the different results. Currently, however, the analogue-based results receive little attention. I recommend devoting more space to this synthesis in a scientific discussion, and reducing the current emphasis on impact attribution, given that impact attribution is not the focus of the study.

Among the scientific comments, see for example, the one on the summary statistics (doubling likelihood from climate change that you also report in the abstract) derived from the probabilistic approach.

The paper's writing style is quite unique, featuring multiple subsections in the Introduction (quite unusual). The rest (after the introduction) is composed of individual sections that combine methods and results for each approach. Although unconventional, this macro-structure after the introduction suits the paper in my view. However, each approach-specific section requires a clear separation between Methods and Results. This divide is missing in the Analogues and IRIS sections, but is present for the probabilistic approach, creating an inconsistency and making the reading sometimes uneasy. For instance, the Analogues section continuously blends methods and results.

Overall, while the reasoning of the text is generally sound, the manuscript would benefit from more conciseness. It is key to more directly focus on the main point of interest.

Figures/captions are not up to the standard for a scientific publication. Please also see my comments in a specific section at the end of the file, and other comments before that section too. These require particular attention to improve precision and overall quality, as many aspects must be guessed. Furthermore, I recommend merging several figures into a single multi-panel figure to help readers avoid getting lost and to present closely related panels together.

Overall, the concept of the paper (using multiple methods for attribution) is interesting and fits NHESS. However, the presentation of the paper needs large improvements. Please address the specific comments below — including relevant scientific aspects — and the technical comments.

**Specific comments**

**Abstract**

Final sentence on the damage: Given you do not carry an impact assessment, I would rather use something weaker here, such as "these results indicate/suggest that climate change played a key role in amplifying the damages…"

L19, mention results for the analogue-based analysis.

**Introduction**

The introduction is lengthy. The author could condense the text where possible to enhance readability. As a very interesting aspect of this study is the usage of multiple approaches, it is worth succinctly highlighting that using different approaches is relevant and why, just before the "This study" subsection.

L33, add year

L68 asserts there is uncertainty, while L74 makes a bold statement, solve the inconsistency.

L77, I am not fully convinced, as there is quite some debate on the cited Nature paper (see e.g. the "matters arising" associated with Kossin's article).

L114, is it really necessary to use the abbreviation NA? Avoid if not used often.

L151, grammar, that what? Also, the percentage has an uncertainty, but the FAR not, so the "equating" does not work.

L153, the - should be longer —.

For example, about repetitions in the text, around L175, there are some repetitions. Shortening this may also allow the authors to avoid the summary at line 191 (another repetition), which would shorten things further.

**Circulation analogues**

The section is a blend of methods and results. It is better to have a mini Methods section and a mini Results section to help the reader (the same applies to the IRIS section later on).

In general, please refer to some of the previous literature that used a similar analogue-based approach. (By the way, for the next section on probabilistic approach, many references are provided, creating another inconsistency.)

L220, do you mean you focus on analogues that resemble the day on which the extreme unground happened?

L220 (around here), What metric (distance) do you use to identify analogs? Make it clear before the results are presented.

How do you deal with the resolution gap between models and "observation-based reanalysis ERA5"?

The selection of the models appears somewhat subjective or not entirely clear. I appreciate using high-resolution models, but, for example, it seems that the discarded HadGEM3-GC31 low-resolution model might appear better than other selected high-resolution models.

The caption of Figure 4 suggests that you use the "highest Spatial correlation" therefore the day with the highest similarity to the observed event for reassessing the goodness of the model in reproducing analogues of Odette. But then, for the attribution assessment, you use the 5 days with the highest similarity. This creates some inconsistency, as among the top 5 days, only one may be similar to Odette.

L301, I appreciate stating that the SD is only for illustrative purposes given the small sample size (5). Shall you just use the full range across events instead? Figure 6 (clarify what the error bar is in the caption).

L305 "EC-Earth3P-HR is the only model showing a slight decrease in daily precipitation from 1950-1970 to 2001-2021." This may indeed be related to the very small sample size and the associated displacement of rainfall extremes in the analogues. Discuss.

L327, does it mean that you also use high-resolution models in the next, probabilistic section? Please clarify succinctly by adding a few words. If not, isn't this a problem for the next, probabilistic section, and to what extent? Discuss if needed, where appropriate in the text.

Caption of Figure 6 and L300, either I missed it, or it is not stated over which domain this rainfall/wind is taken or aggregated? Clarify.

**Probabilistic approach**

L340, I am puzzled here as, while it is not clear to me which spatial box was used in the previous section for Figure 6 (fix there please!), here — given that you re-clarify the box used — it seems you use a different one.

If the box is different, please ensure that you use the same standard (spatial box) for all analyses to ensure comparability and a final synthesis of the results based on the multiple methods.

If the box is the same, then this should be defined in a common instance before for all sections.

L399, "with MSWEP being a statistically significant estimate", I understand it, but in reality, this current wording has no meaning. Improve. Furthermore, the others are not significant? Also, state at what level (significance).

L425, Succinctly clarify to the reader why models are needed. This is crucial for the reader to accurately interpret the net results. In principle, if one can derive observation-based estimates, one does not need models. Here, I think there is a matter of internal variability and sample size that somehow makes the observation-based estimates partially unreliable for the purpose of disentangling human-induced climate change, but it is not explained and remains unclear to people who are not familiar with WWA practice. Clarify this scientific aspect.

L431-440, this is again a sort of explicit blending of Methods and Results (even within this section (probabilistic approach) where you separate methods and results).

L447, for the intensity part, does it mean that once the evaluation is passed, then the absolute values of the rainfall in the models (e.g., in mm) are trusted, and the thresholding for the PR is done via the observed value of the rainfall event (in mm) from ERA5?

Furthermore, is the likelihood you refer to based on the GEV fit? Clarify. In general, make sure that no aspects that are obvious to WWA readers are not clear to non-WWA readers. This is a self-standing scientific publication and scientific standards need to be met.

L452, add an explanation of how synthesis is obtained for observations. The synthesis of the models is also not explained clearly enough.

In general, about the synthesis of probability ratios PRs: if one averages PR (that should be in between 0 and 1 for negative climate change effect, and in between 1 and infinity for positive climate change effect), isn't the average giving more weight to values above 1 with positive climate change effect? I assume that the weighting is accounting for this effect, as otherwise this would be problematic. Please clarify in the text explicitly.

L455, some of this info can go in the caption for the figures.

L484 "more likely than not" in IPCC language has a precise meaning. Here, it does not seem you can make a quantitative judgment on probabilities given the discrepancies. I appreciate how difficult it is, but precision and accuracy are needed.

L496, cite the other attribution studies.

It seems hard to make a simple likelihood statement—as you note around L502 and given the systematic model-observation discrepancy for the December case. Yet at L500 you report a clear figure (a doubling of likelihood, stated also in the abstract) based on June-December statistics. You then invoke physical understanding, which is, however, directly relevant to event magnitude under specific circulation patterns (cyclone), but not directly to likelihood (i.e., how often such cyclones occur), which remains more uncertain. Can you resolve this apparent contradiction? If you give a number, ensure it is well grounded.

**IRIS**

L531, "Observed historical TC tracks between 1980 and 2021 from the IBTrACS database are stochastically perturbed by the model." Does it mean that the approach is still conditional, that is, build an ensemble by assessing how the given observed cyclone tracks could have tracked differently (in space and intensity), but not whether different cyclones than those in IBTrACS could have formed due to large-scale internal varibaility? Please clarify, including how this affects the interpretation of the attribution statements (probability ratio).

Partially related to the above, you seem to use the observed PI trend since 1979 from ERA-5 for anchoring your cyclone model. How is not considering internal climate variability driven uncertainty in this trend (as you use only the observed one, which is a single realisation of the climate) affecting the final results? Similar considerations apply for anchoring on the IBTrACS (see comment above).

L551, "The anthropogenic trend is assumed to be the global zonal mean PI trend, removing potential model biases and regional-scale variability, and use the observed PI trend since 1979 from ERA-5." Unclear, which models? I guess you mean that here you do not have biases compared to approaches that use models. This would align with the next sentence, "This approach avoids the need for any specific climate model and is therefore both simple and robust." Please clarify.

L553-556, this part is not completely clear to me (again, I can guess, but that's not what I should do here). Please create a comprehensive Method section that includes all relevant aspects. Accordingly, combine figures 10 and 11, and clarify the caption (relevant for following the methods correctly).

Caption Figure 13, clarify the caption by referring to the bold line first explicitly; this confused me initially. Note in the caption also that (I guess…) a fair comparison between the uncertainty range from IRIS (factual) and the IBTrACS is not really possible because in IRIS you fix the warming level while IBTrACS takes multiple years over different warming levels.

L589, "This result agrees with other attribution analysis for typhoons in the region using different approaches". Interesting, succinctly mention what approach they used. For example, high-resolution model based, or observations based etc

L604, I appreciate the uncertainty provided by obtaining multiple ensemble members, but I would like to know why you use 120 years per member.

Also, is this obtained for the all-Philippines case? Clarify, also in the caption of Figure 15.

**Final part**

L635, where is the range taken from? Clarify.

L614, it seems you mean that you now derive the FAR also for section 3, so you can compare sections 3 and 4. Please make the sentence more direct.

You bring together three different approaches, which is a strength of the paper; however, the analogue-based approach is not thoroughly considered in the final discussion, where instead a reader is expected to see a comprehensive discussion that integrates the different results. (The analogs-based section is not supposed to serve only as an evaluation section.)

L642-674, I wonder how relevant this part is, given that the study clearly does not attribute any impacts, and the focus is rather on the hazard. Making statements about impacts is a leap, and it is based on potentially important underlying assumptions, including the "event definition". For example, at L714, you refer to at least US$450 million of attributable damage (by the way, where is this number referred to in the previous section?). I would suggest removing this part and rather

focusing on discussing the results about the hazard, as done for example in the final part of the discussion.

L707, gives -> suggests

**Technical comments**

**Circulations analogues: Technical comments**

L220, Sect 3 and Sect 4, this reference to sect 3 and sect 4 is somewhat misleading.

Consider moving this sentence to the next paragraph (which partially addresses my questions) and avoiding unnecessary repetitions.

L285, great, though it could be shortened by reducing repetitions.

L303, long hyphen needed here and in other locations of the paper

L328 When referring to "this section," it's better to say "the current section" to avoid misunderstandings.

**Probabilistic approach: Technical comments**

L378, define PEA

L349, 70-80% is demonstrated in section 3.2.1? If so, move the brackets, otherwise support with reference.

Table 2, I can imagine well what the values are, but the reader should not imagine. Please improve the caption so all is precisely defined, define better "such an event" (in terms of return period). Furthermore, some blue is darker than others, explain.

Table 3 has no colors, hinting at significance as relevant for the colour, but then different colours are in Table 2. Clarify

I suggest combining Fig 8 and Table 4, and Figure 8 and table 5. Then, avoid repeating caption when things are the same as in another figure with a single exception, use "The same as…but …"

Table 4 and table 5, ensure the colours and could darkness is correct and properly applies to the tables. Are you using a different colour scheme (bold vs colour) in the two tables? If so, adjust using a common approach.

**IRIS: Technical comments**

Are you citing all Figures 10-15? If not, fix.

Please combine some of the figures of this section, e.g 10-11 and 12-13-14

L536, "As a third method" it seems this paragraph was written without considering the previous one, you are still taking about the same IRIS approach, are you? Improve text and avoid unnecessary repetitions.

552 PI, either you use it or not, sometimes you have PI , other potential intensities, creating confusion.

L596, refer to Figure 14

**Specific comments on Figures**

Note that some comments are reported for a single figure only below, but the authors should verify if they also apply to other figures, as this is often the case and I did not want to repeat the comments multiple times.

Figure 1, it is generally recommended to use a discrete palette as you did in Figure 4

Figure 2, green/red, ensure it is color-blind-friendly. Talk of box. Cyclones for which attribution studies exist (the authors can consider adding "to our knowledge")

Figure 4, caption: grammar, the second comma is misplaced.

Figure 5, grammar caption. The text is very small and not readable. Adjust to the standard, if needed, also in other figures.

Figure 4,5,6, Regarding the figure quality (I mean the file resolution), it seems to me very low and should be improved.

Figure 5, caption, use semicolons, punctuation where needed.

Figure 6, specify that the error bar shows (eg range cross 5 analogues).

Figure 7, panel a: Describe the vertical lines on the x axis. Panel b: What is the dashed line? What is the scale of the precipitation, e.g. daily, annual max etc. Clarify

Figure 13 "Return curves", please be rigorous.

Figure 13/14, please combine them into a unique figure so as to reduce the number of figures.

Are you referring to Figure 14 in the text? As I write this specific sentence, from a quick search, it seems not. If so, check the same for all figures. All figures should obviously be mentioned in the text.

15 figures is a lot. In general, I strongly recommend combining multiple figures that belong to the same conceptual analysis if possible (that is, if the resulting figure is not too large) so as to help the reader orient themselves better.

Figure 8/9, combine in a unique figure!